# From Human Labels to Literature: Semi-Supervised Learning of NMR Chemical Shifts at Scale

Yongqi Jin [1 2 *]   Yecheng Wang [1 3 *]   Jun-Jie Wang [2 4]   Rong Zhu [4 3]   Guolin Ke [2]   Weinan E [1 3 5]

## Abstract

Accurate prediction of nuclear magnetic resonance (NMR) chemical shifts is fundamental to spectral analysis and molecular structure elucidation, yet existing machine learning methods rely on limited, labor-intensive atom-assigned datasets. We propose a semi-supervised framework that learns NMR chemical shifts from millions of literature-extracted spectra without explicit atom-level assignments, integrating a small amount of labeled data with large-scale unassigned spectra. We formulate chemical shift prediction from literature spectra as a permutation-invariant set supervision problem, and show that under commonly satisfied conditions on the loss function, optimal bipartite matching reduces to a sorting-based loss, enabling stable large-scale semi-supervised training beyond traditional curated datasets. Our models achieve substantially improved accuracy and robustness over state-of-the-art methods and exhibit stronger generalization on significantly larger and more diverse molecular datasets. Moreover, by incorporating solvent information at scale, our approach captures systematic solvent effects across common NMR solvents for the first time. Overall, our results demonstrate that large-scale unlabeled spectra mined from the literature can serve as a practical and effective data source for training NMR shift models, suggesting a broader role of literature-derived, weakly structured data in data-centric AI for science. Our code is available at https://github.com/YongqiJin/NMRNetplusplus.

---

*Equal contribution [1]School of Mathematical Sciences, Peking University, Beijing, China [2]DP Technology, Beijing, China [3]AI for Science Institute, Beijing, China [4]College of Chemistry and Molecular Engineering, Peking University, Beijing, China [5]Center for Machine Learning Research, Peking University, Beijing, China. Correspondence to: Yongqi Jin <yongqijin@stu.pku.edu.cn>, Weinan E <weinan@pku.edu.cn>.

*Proceedings of the $43^{rd}$ International Conference on Machine Learning*, Seoul, South Korea. PMLR 306, 2026. Copyright 2026 by the author(s).

## 1. Introduction

Nuclear magnetic resonance (NMR) spectroscopy is a cornerstone technique for molecular structure elucidation, providing detailed information on the chemical environments of individual atoms (Clayden et al., 2012; Skoog et al., 2019). Among the various NMR observables, chemical shifts are particularly important for assigning spectral peaks and interpreting complex molecular structures (Smith & Goodman, 2010; Chen et al., 2020; Hu & Qiu, 2023; Jin et al., 2025). As a result, accurately predicting chemical shifts is a fundamental task in molecular science and related fields (WISHART, 1994; Tsai et al., 2022). In computational terms, this task can be formulated as: given a molecular structure, predict the chemical shift of each NMR-active nucleus, typically for hydrogen ($^1$H) and carbon ($^{13}$C) nuclei.

While traditional methods such as density functional theory (DFT) (Wolinski et al., 1990) and HOSE codes (Bremser, 1978) provide valuable predictions, they struggle to balance accuracy with computational efficiency. In recent years, machine learning (ML)-based approaches have emerged as powerful alternatives, offering substantial speedups over DFT calculations while achieving high prediction accuracy (Han et al., 2022; Zou et al., 2023; Chen et al., 2024; Xu et al., 2025). However, these methods predominantly rely on manually curated datasets with explicit atom-level assignments, such as NMRShiftDB2 (Kuhn et al., 2012; Kuhn & Schlörer, 2015), which are limited in size and are labor-intensive to generate. Moreover, these datasets fail to capture the effects of solvents on chemical shifts, a critical aspect of real-world NMR spectra (Buckingham et al., 1960).

While the scarcity of human-labeled datasets remains a significant challenge, scientific literature offers vast quantities of NMR spectra, often accompanied by experimental details including solvent information. Recent advancements in document parsing technologies (Wang et al., 2024) and Optical Chemical Structure Recognition (OCSR) (Fang et al., 2024) have made it increasingly feasible to extract large-scale NMR data from literature (Wang et al., 2025b;a). However, these NMR data often lack atom-level assignments, which makes them difficult to incorporate into traditional

supervised learning frameworks. This emerging resource presents an exciting opportunity to apply weakly supervised learning techniques, enabling models to leverage the wealth of unassigned chemical shift data.

To address these challenges, we introduce a weakly supervised learning that leverages unlabeled literature data during training. Additionally, we integrate solvent information as input to capture solvent effects in chemical shift prediction. This framework overcomes the limitations of previous supervised machine learning models, which were constrained by the scalability of labeled experimental data, and achieves superior performance in both prediction accuracy and generalizability.

**Our contributions** can be summarized as follows:

- We formulate NMR chemical shift prediction from unassigned literature spectra as a permutation-invariant set supervision problem, and propose a semi-supervised learning framework that jointly leverages atom-assigned data and large-scale unassigned spectra.

- We show that under mild conditions on the loss function, the optimal bipartite matching between predicted and observed chemical shift sets reduces to a sorting-based loss, enabling stable and scalable training with millions of weakly supervised samples.

- We curate a large-scale (millions of spectra) literature-extracted NMR chemical shift dataset, ShiftDB-Lit, containing molecular structures, chemical shift sets and solvents, covering $^1$H, $^{13}$C, and multiple heteroatoms.

- Using this dataset, we incorporate experimental solvent information into large-scale chemical shift learning, and demonstrate that global solvent conditioning captures systematic solvent-induced biases in NMR prediction. We further provide labeled benchmarks and baselines for heteroatom shifts ($^{19}$F, $^{31}$P, $^{11}$B, $^{29}$Si).

Overall, this work demonstrates that unassigned literature NMR spectra can be transformed into an effective learning signal through permutation-invariant supervision. By leveraging large-scale weakly supervised data together with limited atom-assigned labels, our approach enables more accurate, solvent-aware, and multi-element chemical shift prediction. More broadly, this study highlights the potential of literature-derived datasets—despite their lack of explicit standardization—as a powerful and largely untapped resource for scientific machine learning.

**Conflict of Interest Disclosure** The authors declare no financial conflicts of interest related to this work.

## 2. Related Work

**Chemical Shift Prediction.** Traditional methods primarily rely on empirical induction or quantum mechanical calculations to predict chemical shifts based on atomic environments. Density Functional Theory (DFT) (Wolinski et al., 1990) provides a more accurate quantum mechanical approach, but it is computationally intensive and not feasible for large datasets or high-throughput applications. The HOSE code methods (Bremser, 1978) use predefined rules based on local atomic environments to estimate chemical shifts. While fast, these methods can lack accuracy for complex molecules. With the rise of machine learning, supervised models based on molecular graphs, including GNNs and equivariant MPNNs, have achieved substantially improved accuracy over rule-based methods (Jonas & Kuhn, 2019; Kwon et al., 2020; Han et al., 2022; Zou et al., 2023). More recently, GT-NMR (Chen et al., 2024) and NMR-Net (Xu et al., 2025) adopted graph Transformers (Vaswani et al., 2017) to represent the relationships between atoms in molecules. The latter uses an SE(3)-equivariant Transformer, taking molecular spatial coordinates as input, further enhancing the prediction accuracy of ML methods. Despite architectural advances, all existing ML-based chemical shift predictors fundamentally rely on atom-level assignments during training, implicitly assuming a one-to-one correspondence between atoms and spectral peaks. This assumption breaks down for the vast majority of literature-reported NMR spectra, motivating alternative supervision paradigms.

**NMR Datasets.** The largest experimental chemical shift dataset is NMRShiftDB2 (Kuhn & Schlörer, 2015), which has collected about 40,000 $^1$H and $^{13}$C NMR spectra. QM9-NMR (Gupta et al., 2021) contains over 130,000 molecules with their calculated chemical shifts using DFT methods. With advancements in literature parsing tools, some methods are now attempting to extract large-scale NMR data from scientific literature. NMRBank (Wang et al., 2025b) utilizes a fine-tuned large language model to identify and extract relevant data, while NMRexp (Wang et al., 2025a) employs a pipeline combining PDF parsing, Optical Chemical Structure Recognition (OCSR), and LLM-based extraction methods, obtaining a million-scale dataset of unassigned molecular NMR data. While literature-extracted datasets are orders of magnitude larger, their lack of atom-level correspondence renders them incompatible with standard supervised objectives used in existing models.

**Semi-Supervised and Weakly Supervised Learning.**

Limited labeled data, high acquisition costs, and annotation difficulty are common challenges across many domains.

---

[1]The quantity reports the number of molecules.

[2]The "assigned" annotation for $^{19}$F, $^{31}$P, $^{11}$B, and $^{29}$Si is due to each molecule containing only one equivalent NMR-detectable nucleus (e.g., $^{19}$F) and a single chemical shift in the spectrum.

*Table 1.* Comparison of Datasets

| Dataset | Nuclei | Annotation Type | Solvent Included | Data Quantity[1] |
|---------|--------|-----------------|------------------|---------------|
| NMRShiftDB2 | $^1$H | Assigned | No | 12800 |
|  | $^{13}$C | Assigned |  | 26859 |
| ShiftDB-Lit (Ours) | $^1$H | Unassigned | Yes | 898422 |
|  | $^{13}$C | Unassigned |  | 704373 |
|  | $^{19}$F | Assigned[2] |  | 126961 |
|  | $^{31}$P | Assigned |  | 26980 |
|  | $^{11}$B | Assigned |  | 12902 |
|  | $^{29}$Si | Assigned |  | 1785 |

Semi-supervised and weakly supervised learning aim to leverage unlabeled or weakly labeled data to improve model performance (Van Engelen & Hoos, 2020; Zhou, 2018). In scientific domains, the scarcity of high-quality, standardized data has motivated the use of weakly annotated or unlabeled datasets to train models effectively. Such methods have been explored in tasks including Quantitative Structure-Activity Relationship (QSAR) modeling (Chen et al., 2021; Kwon et al., 2022), synthesis procedure classification (Huo et al., 2019), and drug fingerprint identification (Shi et al., 2023).

**Incorporating Solvent Information for Prediction.** Traditional approaches include explicit and implicit solvent models. Explicit models (Mark & Nilsson, 2001) represent each solvent molecule and use molecular dynamics (MD) simulations to capture solute–solvent interactions accurately, but at high computational cost. Implicit models, such as PCM (Mennucci, 2012) and COSMO (Klamt, 2011), treat the solvent as a continuous medium, reducing computational demands while still requiring substantial resources. In machine learning, solvent information is typically encoded as descriptors (Li et al., 2025). However, in NMR chemical shift prediction, incorporating solvent information remains largely unexplored due to the scarcity of standardized datasets with solvent labels.

## 3. Methods

### 3.1. Data Source and Preprocessing

We constructed our chemical shift dataset from NMR-exp (Wang et al., 2025a), a large-scale collection of NMR spectra extracted from peer-reviewed chemistry publications. The dataset comprises millions of molecular-spectral entries, each linking a molecular structure with its reported NMR spectra. Raw textual NMR data were parsed via regular expressions and transformed into sets of chemical shifts, represented as multisets to capture all observed peaks.

To ensure data quality, we applied three-stage filtering: **molecular validity checks**, **NMR data validity checks**, and **consistency checks**. Molecular validity checks removed

chemically incorrect structures, including free radicals, isotopes, invalid SMILES, or uncommon elemental compositions. NMR data validity checks enforced monotonicity, verified chemical shift ranges to account for potential reporting errors in the literature, and filtered out peaks with excessively broad widths indicative of low-resolution or noisy measurements. Consistency checks ensured alignment between the number of atoms and reported chemical shifts, accounting for ambiguities in carbon spectra where peak integration is unavailable. Finally, we generated 3D molecular conformations using RDKit (Landrum et al., 2016) to provide structural information for molecular modeling.

Moreover, the large volume of literature-extracted data enables large-scale training of heteroatom chemical shifts. Since heteroatoms are typically present in small numbers within molecules, we retained only molecules containing a single equivalent heteroatom, whose spectrum contains exactly one shift and can thus be treated as labeled data.

Detailed data processing procedures are provided in Appendix A, and a comparison between our dataset and previous ones is presented in Table 1.

### 3.2. Semi-supervised Training Framework

We jointly train the model using both atom-assigned (labeled) and unassigned (unlabeled) NMR spectra, as illustrated in Figure 1. The overall objective consists of a supervised atom-level loss for labeled data and a weakly-supervised molecule-level loss for unassigned data.

**Supervised atom-level loss.** For a molecule with atom-level assignments, let $\{s_i\}_{i=1}^N$ denote the predicted chemical shifts and $\{\hat{s}_i\}_{i=1}^N$ the corresponding ground-truth labels. The supervised loss is computed directly at the atom level:

$$\mathcal{L}_{\text{atom}} = \sum_{i=1}^{N} l(s_i, \hat{s}_i), \tag{1}$$

where $l(\cdot, \cdot)$ denotes a pointwise regression loss such as MAE or MSE.

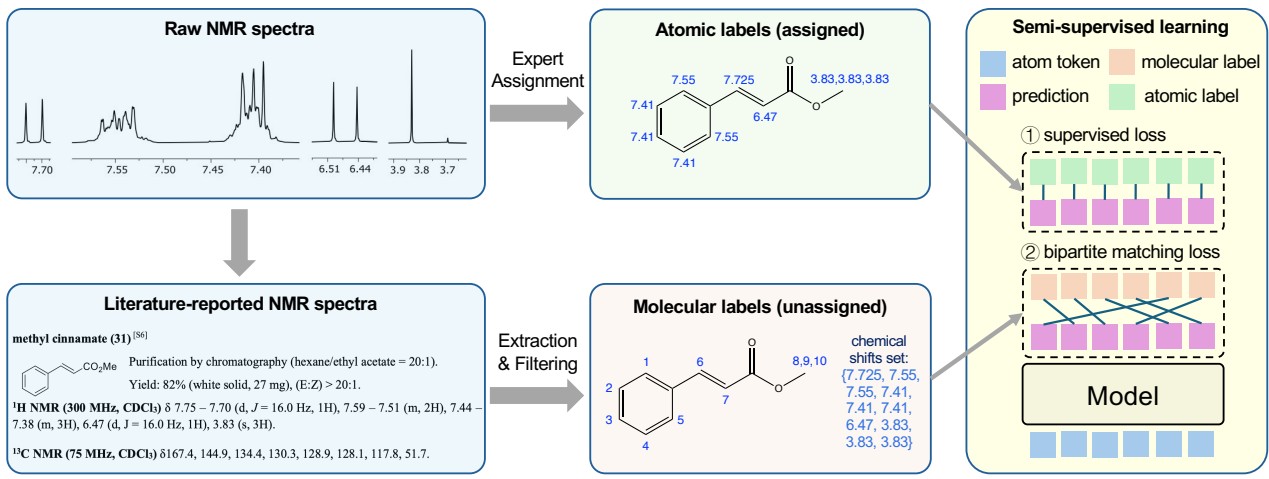

*Figure 1.* Framework of semi-supervised learning for chemical shift prediction models.

**Weakly-supervised molecule-level loss.** For unassigned spectra, the reported chemical shifts form an unordered set and do not correspond directly to specific atoms. We therefore formulate learning from unassigned data as a *permutation-invariant set supervision* problem.

Specifically, we define the **weakly-supervised (molecule-level) loss** as a **bipartite matching loss**, computed as the supervised loss under the optimal assignment between predicted and observed shifts:

$$\mathcal{L}_{\text{mol}} = \min_{\sigma \in \mathfrak{S}_N} \sum_{i=1}^{N} l(s_i, \hat{s}_{\sigma_i}), \qquad (2)$$

where $\mathfrak{S}_N$ denotes the set of all permutations of $N$ elements. In general, this corresponds to solving a bipartite matching problem, which can be computed using the Hungarian algorithm (Kuhn, 1955; Munkres, 1957).

**Permutation-invariant loss equivalence.** When the loss function satisfies

$$l(x, y) = f(|x - y|), \qquad (3)$$

where $f$ is a monotonically increasing and convex function, the optimal assignment admits a closed-form solution. Specifically, the minimum in (2) is achieved by sorting both the predicted and observed shifts, then matching them in ascending order:

$$\mathcal{L}_{\text{mol}} = \sum_{i=1}^{N} l(s_{r_i}, \hat{s}_{t_i}), \qquad (4)$$

where $\{r_i\}$ and $\{t_i\}$ denote the indices that sort $\{s_i\}$ and $\{\hat{s}_i\}$ in ascending order, respectively.

This result transforms a combinatorial matching problem into a deterministic, permutation-invariant loss, enabling stable and efficient training on large-scale unassigned spectra. The above condition holds for commonly used regression losses, including MAE, MSE, and Huber loss. A formal proof is provided in Appendix F.

**Overall objective.** The final training objective combines the supervised and weakly-supervised losses:

$$\mathcal{L}_{\text{total}} = \mathcal{L}_{\text{atom}} + \lambda \, \mathcal{L}_{\text{mol}}, \qquad (5)$$

where $\lambda$ controls the contribution of the weak supervision term.

During the training process, we use batch sizes $B_1$ for supervised and $B_2 > B_1$ for weakly-supervised losses to reduce the variance of the latter. The batch loss is calculated as:

$$\mathcal{L}_{\text{batch}} = \frac{1}{N_1} \sum_{b=1}^{B_1} \mathcal{L}_{\text{atom}}^b + \lambda \cdot \frac{1}{N_2} \sum_{b=1}^{B_2} \mathcal{L}_{\text{mol}}^b, \qquad (6)$$

where $N_1$ and $N_2$ represent the total number of atoms whose chemical shifts are to be predicted in the labeled dataset batch and the unlabeled dataset batch, respectively.

### 3.3. Model Architecture

We adopt NMRNet (Xu et al., 2025) as our baseline, a state-of-the-art deep learning architecture for NMR chemical shift prediction. NMRNet employs an SE(3)-equivariant Transformer to model the spatial relationships among atoms in a molecule, followed by a regression head that outputs each atom's chemical shift. We maintain the original model configuration and training framework to ensure fair comparisons and consistent evaluation across experiments. For additional details, see Appendix C and D.

*Table 2.* Performance comparison of different methods. **Bold** values represent the best results, while underlined values indicate the original best results.

| Method | NMRShiftDB2 ($L_{atom}$) | | NMRShiftDB2 ($L_{mol}$) | | ShiftDB-Lit ($L_{mol}$) | |
|---|---|---|---|---|---|---|
| | MAE | RMSE | MAE | RMSE | MAE | RMSE |
| **$^1$H** | | | | | | |
| HOSE (Bremser, 1978) | 0.3102 | 0.6587 | 0.2771 | 0.5563 | 0.2159 | 0.3931 |
| GCN (Jonas & Kuhn, 2019) | 0.2423 | 0.5491 | 0.2152 | 0.4553 | 0.1712 | 0.3398 |
| FCG (Kwon et al., 2020) | 0.2253 | 0.4914 | 0.2036 | 0.4196 | 0.1562 | 0.2986 |
| SGNN (Han et al., 2022) | 0.2152 | 0.4868 | 0.1915 | 0.4028 | 0.1503 | 0.2943 |
| GT-NMR (Chen et al., 2024)[3] | (0.158) | (0.293) | – | – | – | – |
| NMRNet (Xu et al., 2025) | | | | | | |
| *Baseline* | 0.1972 | 0.4564 | 0.1761 | 0.3896 | 0.1395 | 0.2790 |
| *+ Semi-supervised (Ours)* | **0.1709** | **0.4337** | **0.1492** | **0.3620** | **0.0559** | **0.1846** |
| | (↓ 13.4%) | (↓ 5.0%) | (↓ 15.3%) | (↓ 7.1%) | (↓ 59.9%) | (↓ 33.8%) |
| **$^{13}$C** | | | | | | |
| HOSE (Bremser, 1978) | 2.5804 | 4.8495 | 2.3095 | 4.1902 | 2.3753 | 4.4148 |
| GCN (Jonas & Kuhn, 2019) | 1.3043 | 2.5103 | 1.1573 | 2.1906 | 1.2551 | 2.9506 |
| FCG (Kwon et al., 2020) | 1.3589 | 2.3487 | 1.2190 | 2.0630 | 1.2620 | 2.8902 |
| SGNN (Han et al., 2022) | 1.2606 | 2.2097 | 1.1206 | 1.9138 | 1.2015 | 2.8771 |
| GT-NMR (Chen et al., 2024) | 1.1647 | 2.1434 | 1.0387 | 1.8651 | 1.1077 | 2.8071 |
| NMRNet (Xu et al., 2025) | | | | | | |
| *Baseline* | 1.1518 | 2.1398 | 1.0143 | 1.8513 | 1.2591 | 2.9207 |
| *+ Semi-supervised (Ours)* | **0.9270** | **1.9128** | **0.7765** | **1.5629** | **0.5060** | **2.3494** |
| | (↓ 19.6%) | (↓ 10.5%) | (↓ 23.4%) | (↓ 15.6%) | (↓ 59.8%) | (↓ 19.6%) |

## 3.4. Embedding Solvent Information

Experimental NMR chemical shifts are inherently solvent-dependent. To account for solvent effects, we incorporate learnable solvent embeddings into the model.

The solvent information is encoded as a learnable embedding $e_{solv} \in \mathbb{R}^{d_{model}}$ via a vocabulary-based categorical mapping. Each common solvent (e.g., $CDCl_3$, $DMSO-d_6$) is assigned a unique category index and learns a distinct embedding vector, while all remaining infrequent solvents are grouped into a single "others" category sharing one embedding. This design balances expressiveness for frequent solvents with generalization for rare ones. The resulting solvent embedding is then incorporated into the model via four distinct integration strategies, as illustrated in Figure 2:

**(1) `[CLS]`-token injection.** $e_{solv}$ is added to the `[CLS]` token embedding to provide global solvent context.

**(2) Atom-token pre-backbone injection.** $e_{solv}$ is added to each atom embedding before the backbone.

**(3) Atom-token post-backbone injection.** $e_{solv}$ is added to each atom embedding after the backbone.

**(4) Simple correction.** A learned scalar $b_{solv} \in \mathbb{R}$ is added uniformly to all predicted atomic shifts.

---

[3]The original work predicts only hydrogens bonded to carbon, which is not directly comparable to the full evaluation.

## 4. Experiments

### 4.1. Implementation Details

The ShiftDB-Lit dataset was partitioned into training and test sets with a 4:1 ratio, using a random split. For NMR-ShiftDB2, we follow its pre-defined benchmark split (Kuhn & Schlörer, 2015) to ensure fair comparison with prior work. The models were trained using the same configuration and hyperparameters as those in the original NMRNet implementation, with detailed information provided in Table 8. Each method was evaluated based on Mean Absolute Error (MAE) and Root Mean Squared Error (RMSE) as performance metrics. All experiments were conducted on an NVIDIA RTX 4090 GPU.

### 4.2. Overall Performance Comparison

In this section, we compare the performance of a range of chemical-shift prediction methods, including traditional approaches (HOSE (Bremser, 1978)), machine-learning models (GCN (Jonas & Kuhn, 2019), FCG (Kwon et al., 2020), SGNN (Han et al., 2022), GT-NMR (Chen et al., 2024)), and the current state-of-the-art model, NMRNet (Xu et al., 2025), evaluated under both supervised and semi-supervised learning paradigms.

We evaluated all methods using MAE and RMSE under both atom-level and molecule-level objectives across the NMR-ShiftDB2 and ShiftDB-Lit datasets (atomic-level metrics

are unavailable for ShiftDB-Lit due to the absence of atom-wise assignments). As shown in Table 2, semi-supervised training with both labeled and unlabeled data significantly improves prediction accuracy over prior supervised models, reducing MAE by 13.4% and 19.6% for $^1$H and $^{13}$C on the expert-annotated NMRShiftDB2 benchmark, effectively leveraging unlabeled data to overcome annotation scarcity.

Notably, on the much larger ShiftDB-Lit dataset, the model achieves substantial improvements, with MAE reductions of 59.9% and 59.8% for $^1$H and $^{13}$C, respectively. The NMRNet Baseline is trained only on NMRShiftDB2, making ShiftDB-Lit an out-of-distribution (OOD) test for the baseline, whereas the semi-supervised model leverages unlabeled ShiftDB-Lit data, making it an in-distribution (ID) test. These results reflect both the benefits of semi-supervised learning and the broader chemical coverage from including ShiftDB-Lit, enhancing robustness and generalization, while purely supervised models remain limited by the coverage of labeled datasets and perform worse when extrapolating to molecules outside the training distribution.

### 4.3. Incorporation of Solvent Information

**Solvent-injection Strategies.** We evaluate the four solvent-injection strategies introduced in Section 3.4 against the solvent-free baseline. As shown in Figure 2, incorporating solvent information consistently improves prediction accuracy across both nuclei, with the `[CLS]`-token approach performing best, while simple correction yields only limited gains. These results suggest that solvent effects act as a global contextual bias rather than atom-local perturbations, making `[CLS]`-token conditioning better suited to capture solvent-dependent variations.

Comparing the two types of nuclei, the improvement for $^1$H is more pronounced than for $^{13}$C. This aligns with chemical intuition, as $^1$H chemical shifts are more sensitive to solvent-dependent interactions such as hydrogen bonding and local polarity, whereas $^{13}$C shifts are dominated by more localized electronic environments.

**Per-solvent Performance.** We further assess the impact of solvent conditioning across different solvent domains (CDCl$_3$, DMSO$-$d$_6$, and others). For each solvent group, Table 3 reports the MAE and RMSE achieved by solvent-conditioned models in comparison to a solvent-agnostic baseline. Complete results for all solvents are shown in Appendix E.4.

The results indicate that solvent conditioning is particularly important for accurately predicting chemical shifts in less prevalent solvents, such as DMSO$-$d$_6$. By explicitly incorporating solvent information, the model mitigates biases introduced by solvent-agnostic training and substantially improves performance on underrepresented solvent domains,

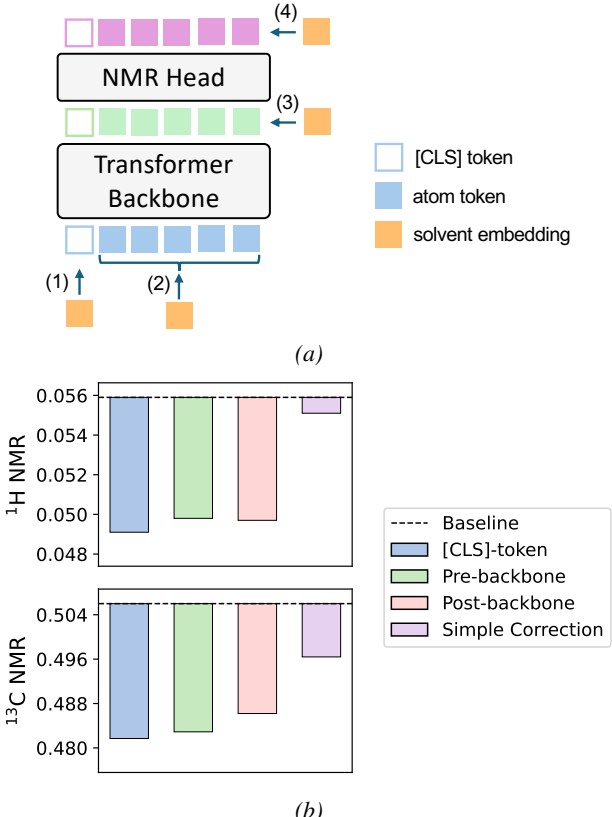

*Figure 2.* **(a)** Strategies to incorporate solvent information into the model: (1) `[CLS]`-token injection; (2) atom-token pre-backbone injection; (3) atom-token post-backbone injection; (4) simple correction (applied after prediction). **(b)** Comparison of these strategies on ShiftDB-Lit in terms of mean absolute error (MAE). The baseline corresponds to the model without solvent information.

especially under highly imbalanced solvent distributions.

We perform cross-solvent validation on molecules in the test set that appear under multiple solvent conditions. For each molecule, predictions are generated using the solvent-conditioned model with the correct solvent token, with incorrect solvent tokens, and without any solvent information. Using the correct solvent token consistently results in the lowest prediction error, while both mismatched solvent inputs and solvent-free predictions lead to higher errors. These results indicate that the model learns solvent-specific adjustments rather than a global chemical shift bias. Detailed results are reported in Appendix E.2.

### 4.4. Heteroatom Chemical Shift Prediction

ShiftDB-Lit provides a rare experimentally labeled dataset for heteroatoms ($^{19}$F, $^{31}$P, $^{11}$B, $^{29}$Si), addressing long-standing data scarcity and enabling standardized benchmarking. Leveraging this dataset, we perform large-scale supervised training and achieve strong predictive accuracy

*Table 3.* Performance comparison across different solvents. Parentheses show relative improvement over the solvent-agnostic model.

| Solvent | Num. Molecules | With Incorporation | | Without Incorporation | |
|---|---|---|---|---|---|
| | | MAE | RMSE | MAE | RMSE |
| $^1$H | | | | | |
| CDCl$_3$ | 162,509 | **0.0475 ($\downarrow$ 5.4%)** | **0.1580 ($\downarrow$ 5.3%)** | 0.0502 | 0.1668 |
| DMSO$-$d$_6$ | 11,623 | **0.0658 ($\downarrow$46.8%)** | **0.2185 ($\downarrow$36.0%)** | 0.1237 | 0.3415 |
| Others | 5,553 | **0.0996 ($\downarrow$ 9.5%)** | **0.2198 ($\downarrow$25.2%)** | 0.1100 | 0.2938 |
| All | 176,985 | **0.0501 ($\downarrow$10.8%)** | **0.1665 ($\downarrow$10.5%)** | 0.0562 | 0.1861 |
| $^{13}$C | | | | | |
| CDCl$_3$ | 126,364 | **0.4775 ($\downarrow$ 2.6%)** | **2.2990 ($\downarrow$ 0.3%)** | 0.4903 | 2.3056 |
| DMSO$-$d$_6$ | 9,026 | **0.6755 ($\downarrow$17.8%)** | **2.5298 ($\downarrow$ 2.0%)** | 0.8223 | 2.5818 |
| Others | 5,485 | **0.8684 ($\downarrow$ 9.0%)** | **3.0113 ($\downarrow$ 1.4%)** | 0.9547 | 3.0530 |
| All | 140,875 | **0.5042 ($\downarrow$ 4.5%)** | **2.3440 ($\downarrow$ 0.2%)** | 0.5281 | 2.3494 |

across all heteroatoms (Table 4). These results establish a solid baseline for heteroatom chemical shift prediction and support future methodological advances.

*Table 4.* Evaluation metrics for heteroatom chemical shift prediction on ShiftDB-Lit.

| Heteroatom | MAE (ppm) | RMSE (ppm) | $R^2$ |
|---|---|---|---|
| $^{19}$F | 2.2809 | 8.7596 | 0.7216 |
| $^{31}$P | 1.3099 | 4.6877 | 0.9634 |
| $^{11}$B | 0.8287 | 2.8560 | 0.9406 |
| $^{29}$Si | 1.9186 | 5.2337 | 0.8901 |

### 4.5. Ablation Study

**Supervised and Weakly-supervised Datasets.** Table 5 summarizes the effects of different combinations of supervised (NMRShiftDB2 with $L_{\text{atom}}$) and weakly-supervised (ShiftDB-Lit or NMRShiftDB2 with $L_{\text{mol}}$) training. Three key observations emerge.

First, augmenting supervised training with an additional weakly-supervised loss does not improve performance over purely supervised learning on both $L_{\text{atom}}$ and $L_{\text{mol}}$ metric (Exp. 1 vs 4). Moreover, training with weak supervision alone leads to a clear degradation in accuracy (Exp. 3), indicating that, when trained on the same fully labeled dataset, augmenting atom-level supervision with additional molecular-level weak supervision does not provide complementary information beyond standard supervised learning.

Second, combining supervised training on the high-quality NMRShiftDB2 labels with weakly-supervised learning on the literature-scale ShiftDB-Lit dataset yields consistent improvements for both $^1$H and $^{13}$C predictions (Exp. 5). These gains persist on the NMRShiftDB2 benchmark despite the substantial distribution shift between the two datasets, suggesting that performance is primarily limited by data scarcity

rather than model capacity. In this regime, weak molecular-level supervision signals distilled from millions of literature spectra act as an effective regularizer.

Third, training exclusively on the weakly-supervised ShiftDB-Lit dataset leads to model collapse (Exp. 2), revealing an inherent failure mode of permutation-based weak supervision. Without atom-level anchoring, early prediction errors are amplified through incorrect bipartite matching, resulting in erroneous atom–peak associations. This effect manifests as substantially worse $L_{\text{atom}}$ metrics compared to $L_{\text{mol}}$, and also hampers training convergence. For example, in $^{13}$C prediction, models trained solely on ShiftDB-Lit underperform the combined NMRShiftDB2 and ShiftDB-Lit setting even when evaluated under $L_{\text{mol}}$.

Overall, these results indicate that weak supervision alone is inadequate, but becomes highly effective when anchored by a moderate amount of high-quality labeled data. The combination of NMRShiftDB2 and ShiftDB-Lit therefore provides the most effective configuration for training robust and accurate NMR chemical shift predictors.

**Weight $\lambda$.** The hyperparameter $\lambda$ controls the trade-off between atom-level supervised learning and molecular-level weak supervision in the training objective. In practice, its effective range reflects a balance between leveraging additional weakly-supervised signals and maintaining training stability under noisy pseudo-labels or weak supervision.

Figure 3 illustrates the model performance under different values of $\lambda$. On the NMRShiftDB2 benchmark evaluated with $L_{\text{atom}}$, we observe a clear U-shaped trend for both $^1$H and $^{13}$C NMR, indicating that both underweighting and overweighting the weakly-supervised component can be detrimental. When $\lambda$ is too small, the model fails to effectively exploit the complementary information provided by weak supervision. Conversely, overly large values of $\lambda$ cause the training objective to be dominated by molecular-level constraints, biasing the optimization toward degenerate

*Table 5.* Ablation study on the effect of the training dataset. "DB2" refers to the NMRShiftDB2 dataset, and "DB-Lit" refers to the ShiftDB-Lit dataset. A dash ("–") indicates that the corresponding dataset or loss is not used.

| No. | Training Datasets | | NMRShiftDB2 ($L_{atom}$) | | NMRShiftDB2 ($L_{mol}$) | | ShiftDB-Lit ($L_{mol}$) | |
|---|---|---|---|---|---|---|---|---|
| | supervised | weakly-supervised | MAE | RMSE | MAE | RMSE | MAE | RMSE |
| $^1$H | | | | | | | | |
| 1 | DB2 | – | 0.1972 | 0.4564 | 0.1761 | 0.3896 | 0.1395 | 0.2790 |
| 2 | – | DB-Lit | 0.2412 | 0.5600 | 0.2103 | 0.4533 | 0.0543 | 0.1849 |
| 3 | – | DB2 | 0.2308 | 0.4902 | 0.1963 | 0.4051 | 0.1439 | 0.2835 |
| 4 | DB2 | DB2 | 0.2152 | 0.4844 | 0.1829 | 0.3968 | 0.1413 | 0.2840 |
| 5 | DB2 | DB-Lit | **0.1709** | **0.4337** | **0.1492** | **0.3620** | **0.0559** | **0.1846** |
| $^{13}$C | | | | | | | | |
| 1 | DB2 | – | 1.1518 | 2.1398 | 1.0143 | 1.8513 | 1.2591 | 2.9207 |
| 2 | – | DB-Lit | 1.5214 | 4.6658 | 1.2931 | 3.1443 | 0.9965 | 2.5962 |
| 3 | – | DB2 | 2.3848 | 4.0615 | 2.1943 | 3.7542 | 2.0393 | 3.9687 |
| 4 | DB2 | DB2 | 1.1503 | 2.2251 | 0.9753 | 1.8541 | 1.1730 | 2.9139 |
| 5 | DB2 | DB-Lit | **0.9270** | **1.9128** | **0.7765** | **1.5629** | **0.5060** | **2.3494** |

solutions that disregard atom-level correctness.

This effect is particularly evident for $^1$H predictions under the ShiftDB-Lit ($L_{mol}$) metric, which generally decreases as $\lambda$ increases. In the absence of sufficient atom-level anchoring, excessive emphasis on weak supervision encourages solutions that optimize molecular-level matching while sacrificing correct atom–peak correspondences. As a result, the molecular-level objective can be increasingly satisfied during training, whereas atom-level accuracy deteriorates, indicating model collapse.

**Training Strategies** To validate the effectiveness of our semi-supervised learning approach, we conduct an ablation study comparing different training strategies. We evaluate four configurations: (1) direct semi-supervised training on both NMRShiftDB2 (DB2) and ShiftDB-Lit (DB-Lit), (2) supervised pretraining on DB2 followed by semi-supervised finetuning, (3) semi-supervised pretraining followed by supervised finetuning, and (4) weakly-supervised pretraining on DB-Lit only, followed by supervised finetuning on DB2.

The results demonstrate two key findings. First, training on large-scale literature spectra without atom-level assignment is effective: even when using only weakly-supervised learning on DB-Lit (Exp. 4), the model achieves competitive performance, validating that our set-level loss can effectively leverage unassigned spectral data. Second, our set-level loss is effective and robust across different training regimes: it performs well both under joint training (Exp. 1) and under pretraining-plus-finetuning strategies (Exp. 2-4). The best overall performance is achieved through direct semi-supervised joint training (Exp. 1) or through strategic combinations of pretraining and finetuning (Exp. 2-3), with different configurations excelling on different evaluation metrics.

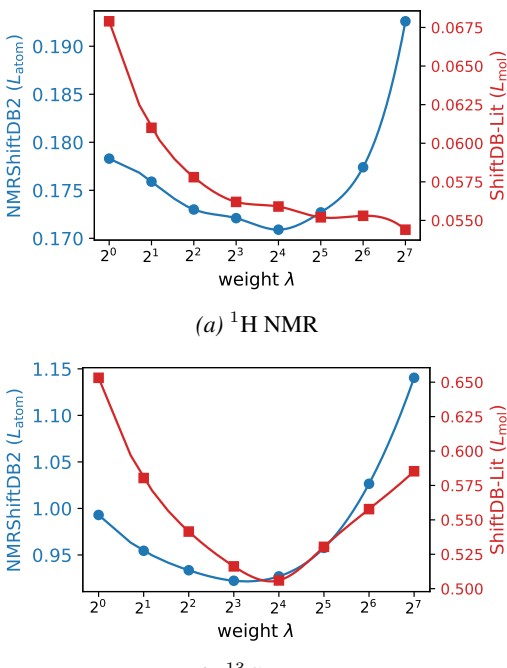

*(a)* $^1$H NMR

*(b)* $^{13}$C NMR

*Figure 3.* Model performance under different values of the weakly-supervised weight $\lambda$ (MAE metric).

## 5. Conclusion

In this work, we address the long-standing data bottleneck in NMR chemical shift prediction by framing learning from literature-derived spectra as a semi-supervised, permutation-invariant problem over unordered supervision signals. By combining a small set of atom-assigned labels with millions of unassigned spectra, we demonstrate that meaningful supervision can be effectively extracted from unordered ex-

*Table 6.* Ablation study on training strategies. "DB2" refers to the NMRShiftDB2 dataset, and "DB-Lit" refers to the ShiftDB-Lit dataset. "Sup." = supervised, "Semi." = semi-supervised, "Weak." = weakly-supervised. A dash ("–") indicates no pretraining or finetuning stage.

| No. | Training Strategy | | NMRShiftDB2 ($L_{atom}$) | | NMRShiftDB2 ($L_{mol}$) | | ShiftDB-Lit ($L_{mol}$) | |
|---|---|---|---|---|---|---|---|---|
| | Pretrain | Finetune | MAE | RMSE | MAE | RMSE | MAE | RMSE |
| $^1$H | | | | | | | | |
| 1 | Semi. (DB2+Lit) | – | **0.1709** | 0.4337 | **0.1492** | 0.3620 | 0.0559 | 0.1846 |
| 2 | Sup. (DB2) | Semi. (DB2+Lit) | 0.1725 | 0.4393 | 0.1503 | 0.3658 | **0.0548** | **0.1838** |
| 3 | Semi. (DB2+Lit) | Sup. (DB2) | 0.1710 | **0.4320** | 0.1496 | **0.3619** | 0.0583 | 0.1857 |
| 4 | Weak. (Lit) | Sup. (DB2) | 0.1744 | 0.4413 | 0.1537 | 0.3657 | 0.0734 | 0.1989 |
| $^{13}$C | | | | | | | | |
| 1 | Semi. (DB2+Lit) | – | 0.9270 | 1.9128 | 0.7765 | 1.5629 | 0.5060 | 2.3494 |
| 2 | Sup. (DB2) | Semi. (DB2+Lit) | 0.9185 | 1.8996 | **0.7675** | **1.5470** | **0.4985** | **2.3424** |
| 3 | Semi. (DB2+Lit) | Sup. (DB2) | **0.9182** | **1.8833** | 0.7779 | 1.5540 | 0.5336 | 2.3570 |
| 4 | Weak. (Lit) | Sup. (DB2) | 0.9345 | 1.8971 | 0.7905 | 1.5620 | 0.6287 | 2.4421 |

perimental observations. This approach offers a scalable solution to chemical shift modeling beyond the constraints of fully labeled datasets.

Our approach leverages a deterministic sorting-based loss for unassigned shifts, enabling stable and scalable training while avoiding the combinatorial complexity of assignment-based objectives. Empirical results demonstrate substantial improvements in prediction accuracy and generalization across diverse molecular structures, solvents, and nuclei. In particular, solvent effects can be captured at scale through simple global conditioning, highlighting the ability of our model to incorporate context-dependent chemical information.

More broadly, this study demonstrates a paradigm shift in scientific machine learning—moving beyond the reliance on small, curated labeled datasets to leverage large-scale, literature-extracted data. Harnessing such vast, unlabeled resources presents a promising path to overcoming the data bottleneck, significantly enhancing model performance in fields where high-quality annotations are scarce. We hope this work will inspire the development of principled methods for learning from weakly structured scientific data, as well as systematic pipelines for extracting and organizing scientific data from the vast literature.

## Impact Statement

This paper presents work whose goal is to advance the field of Machine Learning. There are many potential societal consequences of our work, none of which we feel must be specifically highlighted here.

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

## A. Data Processing

**Molecular Validity Checks**

Due to the presence of many uncommon molecular systems in the literature data, which may affect the stable training and accurate evaluation of the model, we performed a molecular validity check to ensure the quality of the dataset. Structures that met any of the following criteria were filtered out:

- Containing free radicals or isotopes

- Having illegal SMILES (i.e., those that cannot be correctly parsed)

- Uncommon elemental compositions (For C and H spectra, only common elements such as C, H, O, N, S, P, F, and Cl were retained, consistent with the NMRShiftDB2 dataset)

**NMR Data Validity Checks**

Since the literature data may contain reporting errors or typos (e.g., missing symbols or misplaced decimal points), we applied several NMR data validity checks to ensure consistency and quality. These checks included:

- Ensuring the monotonicity of chemical shifts

- Verifying that the chemical shift range was within expected limits

- Filtering out peaks with excessively broad widths, which cannot provide accurate chemical shift labels

**Consistency Check**

To ensure consistency between molecular structures and the corresponding NMR data, we performed a consistency check by comparing the number of atoms in the molecule with the number of chemical shifts observed in the spectrum. For carbon spectra, peak integration information is typically unavailable, making it difficult to directly determine the number of atoms contributing to each resonance. We therefore enforced a consistency criterion requiring that the number of distinct chemical shifts in the $^{13}$C spectrum matches the number of symmetry-unique carbon atoms in the corresponding molecular structure, accounting for possible peak overlap due to molecular symmetry.

**Heteroatom Data Processing**

For heteroatoms, we followed a similar processing workflow to obtain a high-quality labeled dataset:

- **Molecular validity checks:** Similar to the procedures for hydrogen (H) and carbon (C), with the exception that no specific element type constraints are applied.

- **NMR data validity checks:** Chemical shifts must fall within the valid range.

- **Consistency check:** Each molecule contains only one equivalent heteroatom, and the spectrum includes a single chemical shift corresponding to that heteroatom.

*Table 7.* Valid chemical shift ranges for different elements.

| Element | Lower Bound (ppm) | Upper Bound (ppm) |
|---------|-------------------|-------------------|
| $^{1}$H | -1 | 15 |
| $^{13}$C | -10 | 230 |
| $^{19}$F | -300 | 300 |
| $^{31}$P | -150 | 200 |
| $^{11}$B | -50 | 100 |
| $^{29}$Si | -70 | 40 |

### 3D Conformation Generation

We generated molecular conformations using the `EmbedMolecule` and `MMFFOptimizeMolecule` functions from the RDKit toolkit (Landrum et al., 2016), which perform conformational sampling and geometry optimization under the Merck Molecular Force Field (MMFF) (Halgren, 1996).

## B. Dataset Statistics

Figure 4 summarizes the distributions of the number of atoms and chemical shifts per entry in the NMRShiftDB2 and ShiftDB-Lit datasets. Compared to NMRShiftDB2, ShiftDB-Lit not only contains a substantially larger number of entries but also provides a broader coverage of molecules and more complete chemical shift information, highlighting its advantage in scale and diversity.

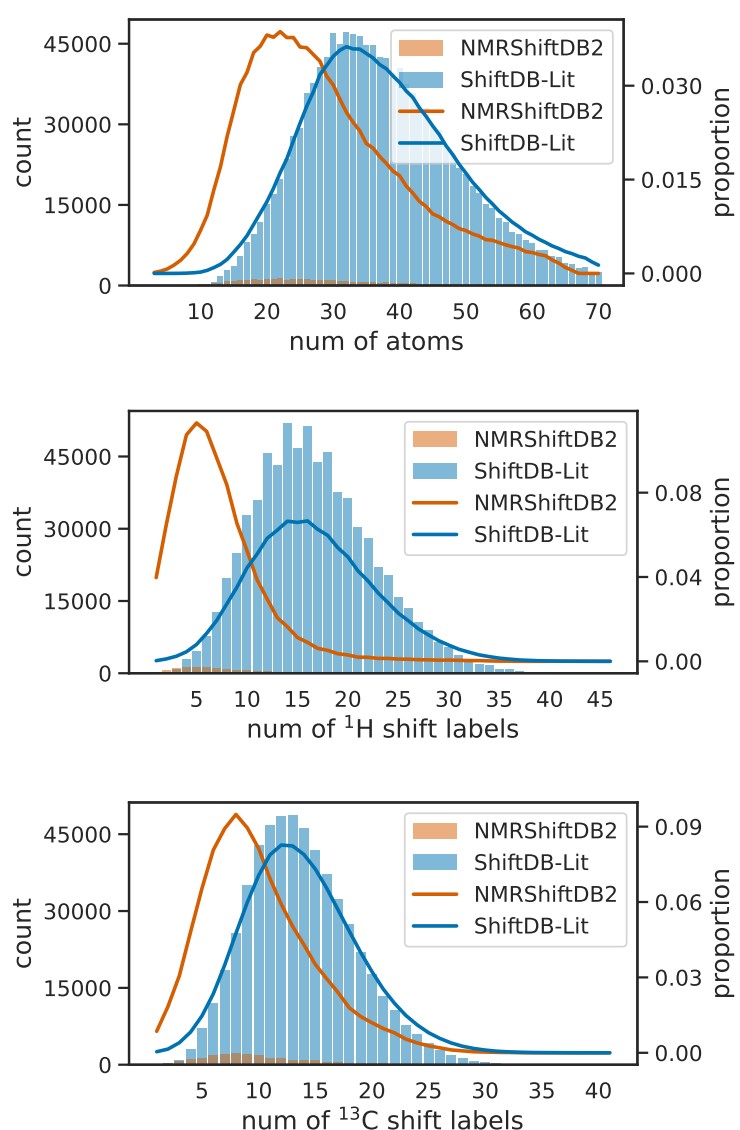

*Figure 4.* Statistics of NMRShiftDB2 and ShiftDB-Lit datasets.

## C. Pretraining and Fine-Tuning Strategy

We adopt a consistent training strategy with the baseline model that combines molecular pretraining with downstream fine-tuning for chemical shift prediction. NMRNet (Xu et al., 2025) leverages pre-trained weights from Uni-Mol (Zhou et al., 2023), obtained via self-supervised learning on a large-scale molecular dataset. During fine-tuning, the model is further trained on the chemical shift dataset to adapt its representations for accurate NMR chemical shift prediction.

## D. Model Architecture and Training Hyperparameter Settings

To ensure a fair comparison and highlight the effectiveness of the semi-supervised approach, we adopt the same model architecture and optimization parameters as those used in the NMRNet model. Unless otherwise specified, this set of parameters is used consistently across all experiments. For both the supervised and semi-supervised settings, training hyperparameters are independently optimized based on validation performance.

*Table 8.* Hyperparameter settings for model architecture and training configuration.

| Hyperparameter | Supervised | Semi-supervised |
|---|---|---|
| *Model Architecture* | | |
| Layers | | 15 |
| Attention Heads | | 64 |
| Embedding Dim | | 512 |
| FFN Hidden Dim | | 2048 |
| Activation Function | | GELU |
| FFN / Attention Dropout | | 0.1 |
| *Optimizer Configuration* | | |
| Optimizer | | Adam |
| Warmup Ratio | | 0.03 |
| Weight Decay | | $1 \times 10^{-4}$ |
| Learning Rate Decay | | Linear |
| Adam's $\epsilon$ | | $1 \times 10^{-6}$ |
| Adam's $(\beta_1, \beta_2)$ | | (0.9, 0.99) |
| Gradient Clip Norm | | 1.0 |
| *Training Hyperparameters (Optimized)* | | |
| Peak Learning Rate | 1e-4 | 4e-4 |
| Batch Size (Labeled Dataset) | 8 | 4 |
| Batch Size (Unlabeled Dataset) | 0 | 16 |
| Epochs | 50 | 10 |
| Weight $\lambda$ | – | 16 |

## E. Expanded Results

### E.1. Different Solvent-injection Strategies

*Table 9.* Evaluations for different mechanisms of solvent incorporation.

| Mechanisms | $^1$H | | $^{13}$C | |
|---|---|---|---|---|
| | MAE | RMSE | MAE | RMSE |
| Without Injection | 0.0559 | 0.1846 | 0.5060 | 2.3494 |
| CLS-token Injection | 0.0491 | 0.1660 | 0.4817 | 2.3373 |
| Pre-backbone Injection | 0.0498 | 0.1664 | 0.4829 | 2.3394 |
| Post-backbone Injection | 0.0497 | 0.1663 | 0.4862 | 2.3377 |
| Simple Correction | 0.0551 | 0.1843 | 0.4964 | 2.3448 |

## E.2. Solvent Pairs Validation

*Table 10.* Cross validation results for molecules in the test set that appear under different solvent conditions.

| Solvent Pairs (Correct / Incorrect) | Num. | Correct Incorporation | | Incorrect Incorporation | | No Incorporation | |
|---|---|---|---|---|---|---|---|
| | | MAE | RMSE | MAE | RMSE | MAE | RMSE |
| $^1$H | | | | | | | |
| $CDCl_3$ / $DMSO-d_6$ | 419 | 0.0832 | 0.4023 | 0.2814 | 0.6976 | 0.1228 | 0.4813 |
| $DMSO-d_6$ / $CDCl_3$ | 419 | 0.0693 | 0.3262 | 0.2815 | 0.6855 | 0.2174 | 0.5919 |
| $CDCl_3$ / Others | 389 | 0.0502 | 0.2368 | 0.0689 | 0.2724 | 0.0567 | 0.2616 |
| Others / $CDCl_3$ | 389 | 0.0871 | 0.2686 | 0.0926 | 0.2856 | 0.0921 | 0.2850 |
| $DMSO-d_6$ / Others | 50 | 0.0326 | 0.0675 | 0.2026 | 0.4761 | 0.1775 | 0.4704 |
| Others / $DMSO-d_6$ | 50 | 0.1160 | 0.2925 | 0.2463 | 0.5439 | 0.1727 | 0.4425 |
| $^{13}$C | | | | | | | |
| $CDCl_3$ / $DMSO-d_6$ | 326 | 0.3432 | 1.3649 | 0.8286 | 1.7025 | 0.4149 | 1.4474 |
| $DMSO-d_6$ / $CDCl_3$ | 326 | 0.4510 | 1.4044 | 0.8501 | 1.6769 | 0.7434 | 1.5806 |
| $CDCl_3$ / Others | 259 | 0.3622 | 2.3394 | 0.5298 | 2.4386 | 0.3994 | 2.3993 |
| Others / $CDCl_3$ | 259 | 0.5032 | 1.2410 | 0.5370 | 1.2955 | 0.6525 | 1.0994 |
| $DMSO-d_6$ / Others | 61 | 0.3518 | 0.7595 | 0.8940 | 1.1805 | 0.6525 | 1.0994 |
| Others / $DMSO-d_6$ | 61 | 0.5214 | 1.0162 | 0.9977 | 1.4678 | 0.7248 | 1.2192 |

## E.3. Per-solvent Performance

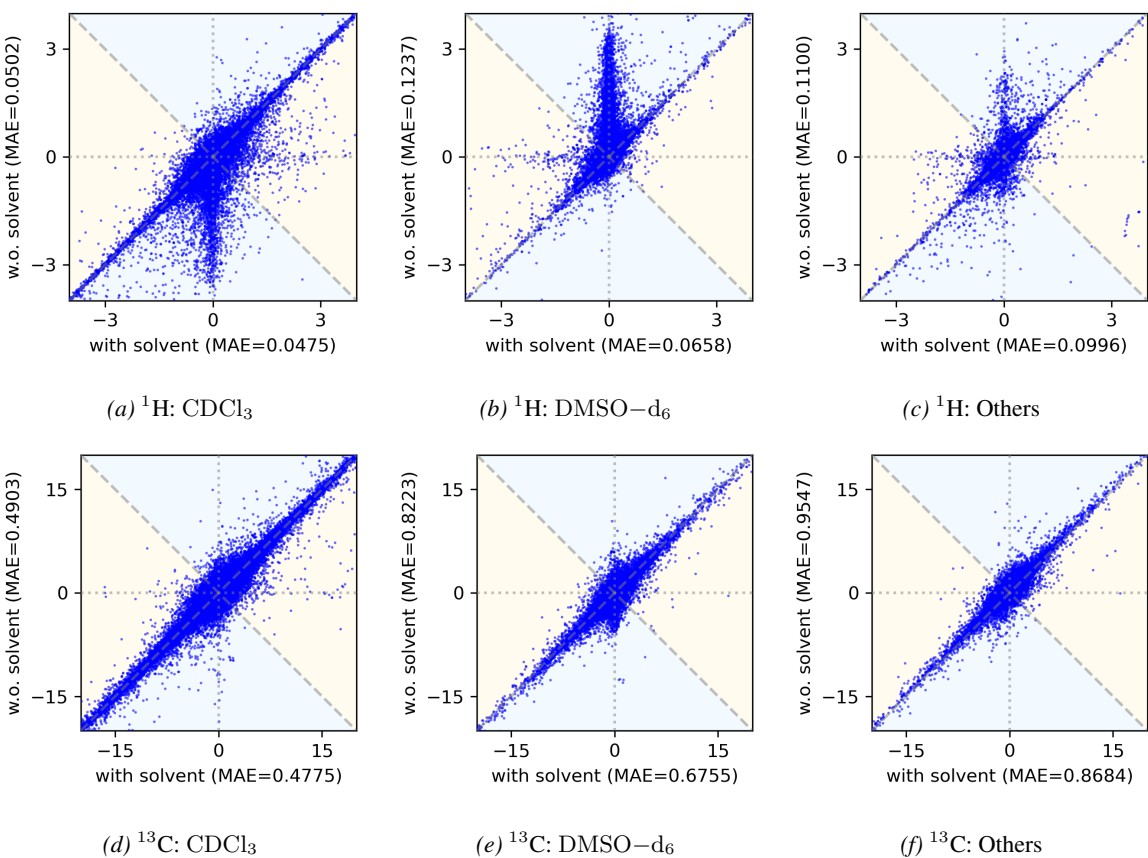

*(a)* $^1$H: $CDCl_3$      *(b)* $^1$H: $DMSO-d_6$      *(c)* $^1$H: Others

*(d)* $^{13}$C: $CDCl_3$      *(e)* $^{13}$C: $DMSO-d_6$      *(f)* $^{13}$C: Others

*Figure 5.* Atom-wise prediction deviations for $^1$H and $^{13}$C NMR shifts under different solvent conditions. The two shaded blue regions indicate where solvent conditioning yields lower errors than the solvent-agnostic baseline.

## E.4. Complete results for all solvents

The magnitude of improvement from solvent incorporation varies systematically across different solvents, reflecting both dataset composition and chemical principles. $CDCl_3$ exhibits relatively modest gains (5.4% for $^1H$, 2.7% for $^{13}C$) because it dominates the training data (89.1% of molecules). Consequently, models without explicit solvent encoding implicitly learn a strong bias toward $CDCl_3$-like environments, leaving limited room for further improvement when solvent information is explicitly provided.

For the more sensitive $^1H$ nucleus, the improvement pattern aligns with chemical intuition regarding solute–solvent interactions. Nonpolar chlorinated solvents structurally similar to $CDCl_3$—such as $CD_2Cl_2$ (9.7% improvement) and $C_2D_2Cl_4$ (1.3% improvement)—show relatively small gains, as their weak intermolecular interactions produce chemical shift perturbations similar to $CDCl_3$. In contrast, polar or hydrogen-bonding solvents including $DMSO-d_6$ (46.8%), $CD_3COCD_3$ (46.0%), $CD_3OD$ (25.6%), $THF-d_8$ (18.9%), and $DMF-d_7$ (29.6%), as well as aromatic solvents such as $C_6D_6$ (59.6%) and $PhMe-d_8$ (48.0%), exhibit substantially larger improvements. This trend is consistent with the expectation that stronger and more specific solute–solvent interactions—including hydrogen bonding, dipole–dipole interactions, and $\pi-\pi$ stacking—more significantly perturb proton chemical shifts, making explicit solvent information more valuable for accurate prediction.

For $^{13}C$, the improvements are generally smaller across all solvents, reflecting the lower sensitivity of carbon chemical shifts to environmental effects compared to proton shifts. Nevertheless, the same trend holds: polar solvents ($DMSO-d_6$: 17.6%, $CD_3COCD_3$: 22.3%, $CD_3OD$: 23.3%) show larger gains than nonpolar ones.

*Table 11.* Complete performance comparison across different solvents. Parentheses show relative improvement over the solvent-agnostic model.

| Solvent | Num. Molecules | With Incorporation | | Without Incorporation | |
|---|---|---|---|---|---|
| | | **MAE** | **RMSE** | **MAE** | **RMSE** |
| $^1H$ | | | | | |
| $CDCl_3$ | 162,509 | **0.0474 ($\downarrow$ 5.4%)** | **0.1578 ($\downarrow$ 5.3%)** | 0.0502 | 0.1668 |
| $DMSO-d_6$ | 11,623 | **0.0654 ($\downarrow$46.8%)** | **0.2194 ($\downarrow$36.0%)** | 0.1237 | 0.3415 |
| $CD_3COCD_3$ | 1,263 | **0.0659 ($\downarrow$46.0%)** | **0.2047 ($\downarrow$34.5%)** | 0.1221 | 0.3125 |
| $CD_2Cl_2$ | 1,030 | **0.0539 ($\downarrow$ 9.7%)** | **0.1269 ($\downarrow$12.2%)** | 0.0597 | 0.1445 |
| $C_6D_6$ | 589 | **0.0927 ($\downarrow$59.6%)** | **0.1916 ($\downarrow$41.7%)** | 0.2294 | 0.3286 |
| $CD_3CN$ | 422 | **0.0794 ($\downarrow$28.8%)** | **0.2520 ($\downarrow$12.8%)** | 0.1115 | 0.2889 |
| $CD_3OD$ | 253 | **0.1685 ($\downarrow$25.6%)** | **0.4767 ($\downarrow$27.1%)** | 0.2266 | 0.6539 |
| $THF-d_8$ | 38 | **0.1243 ($\downarrow$18.9%)** | **0.2517 ($\downarrow$20.6%)** | 0.1532 | 0.3169 |
| $D_2O$ | 34 | **0.5487 ($\downarrow$ 3.2%)** | **1.2667 ($\downarrow$ 4.0%)** | 0.5667 | 1.3189 |
| $DMF-d_7$ | 20 | **0.0825 ($\downarrow$29.6%)** | **0.1153 ($\downarrow$31.4%)** | 0.1172 | 0.1680 |
| $C_2D_2Cl_4$ | 18 | **0.1099 ($\downarrow$ 1.3%)** | **0.2739 ($\downarrow$ 1.0%)** | 0.1114 | 0.2766 |
| $PhMe-d_8$ | 17 | **0.1148 ($\downarrow$48.0%)** | **0.2814 ($\downarrow$16.1%)** | 0.2209 | 0.3355 |
| $CF_3CO_2D$ | 11 | **0.4005 ($\downarrow$ 8.7%)** | **0.5560 ($\downarrow$ 1.6%)** | 0.4389 | 0.5653 |
| pyridine-d5 | 6 | **0.2323 ($\downarrow$ 3.0%)** | **0.3287 ($\downarrow$ -2.2%)** | 0.2395 | 0.3215 |
| Not known | 1,852 | **0.0616 ($\downarrow$ 1.1%)** | **0.1971 ($\downarrow$ 1.6%)** | 0.0623 | 0.2004 |
| All | 176,985 | **0.0492 ($\downarrow$12.5%)** | **0.1646 ($\downarrow$11.6%)** | 0.0562 | 0.1861 |
| $^{13}C$ | | | | | |
| $CDCl_3$ | 126,364 | **0.4772 ($\downarrow$ 2.7%)** | **2.2993 ($\downarrow$ 0.3%)** | 0.4903 | 2.3056 |
| $DMSO-d_6$ | 9,026 | **0.6779 ($\downarrow$17.6%)** | **2.5284 ($\downarrow$ 2.1%)** | 0.8223 | 2.5818 |
| $CD_3COCD_3$ | 1,123 | **0.8120 ($\downarrow$22.3%)** | **2.8229 ($\downarrow$ 2.9%)** | 1.0457 | 2.9082 |
| $CD_3OD$ | 1,105 | **1.1102 ($\downarrow$23.3%)** | **3.4000 ($\downarrow$ 2.3%)** | 1.4474 | 3.4938 |
| $CD_2Cl_2$ | 730 | **0.6366 ($\downarrow$11.8%)** | **2.9171 ($\downarrow$ 0.4%)** | 0.7215 | 2.9285 |
| $C_6D_6$ | 424 | **0.8924 ($\downarrow$ 7.5%)** | **3.7441 ($\downarrow$ 1.0%)** | 0.9652 | 3.7830 |
| $CD_3CN$ | 336 | **0.9443 ($\downarrow$21.1%)** | **3.5589 ($\downarrow$ 1.6%)** | 1.1973 | 3.6178 |
| $D_2O$ | 144 | **1.5524 ($\downarrow$ 9.1%)** | **4.5088 ($\downarrow$ 1.5%)** | 1.7078 | 4.5768 |
| $THF-d_8$ | 79 | **0.7869 ($\downarrow$10.5%)** | **2.8088 ($\downarrow$ 1.0%)** | 0.8794 | 2.8368 |
| $C_2D_2Cl_4$ | 42 | **0.6076 ($\downarrow$ 5.5%)** | **2.5653 ($\downarrow$ 0.5%)** | 0.6431 | 2.5778 |
| $PhMe-d_8$ | 26 | **0.8653 ($\downarrow$ 6.9%)** | **3.1476 ($\downarrow$ 1.2%)** | 0.9293 | 3.1854 |
| Not known | 1,870 | **0.5598 ($\downarrow$ 1.8%)** | **2.4088 ($\downarrow$ 0.3%)** | 0.5702 | 2.4160 |
| All | 140,875 | **0.5060 ($\downarrow$ 3.7%)** | **2.3494 ($\downarrow$ 0.3%)** | 0.5254 | 2.3565 |

### E.5. OOD-vs-ID evaluation

To better disentangle the contributions of data coverage and semi-supervised learning, we stratify DB-Lit test molecules by their maximum Tanimoto similarity (2048-bit Morgan fingerprints) to the union of training structures from DB2 and DB-Lit: similar($\geq$0.7), intermediate(0.5-0.7), dissimilar($<$0.5).

Table 12 summarizes the results. On the similar subset, the semi-supervised model gains the most—consistent with narrowing the OOD–ID gap when training better covers the relevant chemical space. On the dissimilar subset, the semi-supervised model still clearly outperforms the supervised baseline, but the improvement is less pronounced than on the similar subset.

*Table 12.* Performance comparison on in-distribution (ID) and out-of-distribution (OOD) test sets. Test molecules are binned by their maximum Tanimoto similarity to the training set. Lower similarity indicates higher distribution shift.

| Method | <0.5 (OOD) | | | [0.5, 0.7) (Moderate) | | | $\geq$0.7 (ID) | | |
|---|---|---|---|---|---|---|---|---|---|
| | $n$ | MAE | RMSE | $n$ | MAE | RMSE | $n$ | MAE | RMSE |
| $^1$H | | | | | | | | | |
| NMRNet (Supervised) | 3,316 | 0.1964 | 0.4265 | 53,554 | 0.1478 | 0.3001 | 122,815 | 0.1332 | 0.2659 |
| NMRNet (Semi-supervised) | 3,316 | **0.1207** | **0.3620** | 53,554 | **0.0669** | **0.2110** | 122,815 | **0.0491** | **0.1656** |
| *Relative improvement* | | ($\downarrow$38.5%) | ($\downarrow$15.1%) | | ($\downarrow$54.7%) | ($\downarrow$29.7%) | | ($\downarrow$63.1%) | ($\downarrow$37.7%) |
| $^{13}$C | | | | | | | | | |
| NMRNet (Supervised) | 3,043 | 2.2113 | 5.2994 | 45,965 | 1.4188 | 3.4229 | 91,867 | 1.1360 | 2.5034 |
| NMRNet (Semi-supervised) | 3,043 | **1.3924** | **4.7814** | 45,965 | **0.6451** | **2.8615** | 91,867 | **0.4032** | **1.9066** |
| *Relative improvement* | | ($\downarrow$37.0%) | ($\downarrow$9.8%) | | ($\downarrow$54.5%) | ($\downarrow$16.4%) | | ($\downarrow$64.5%) | ($\downarrow$23.8%) |

## F. Proofs in Section 3.2

To prove the equality in (4), we first establish the following lemma.

**Lemma F.1** (Monotonicity and Convexity Lemma)**.** *If $x_1 \leq x_2$ and $y_1 \leq y_2$, then for a monotonically increasing and convex function $f(x)$, we have:*

$$f(|x_1 - y_1|) + f(|x_2 - y_2|) \leq f(|x_1 - y_2|) + f(|x_2 - y_1|)$$

*Proof.* We will prove this lemma by considering two cases based on the relationship between $y_1$, $y_2$, and the midpoint of $x_1$ and $x_2$.

**Case I:** $y_1 \leq \frac{x_1 + x_2}{2} \leq y_2$

In this case, we have:

$$|x_1 - y_1| \leq |x_2 - y_1| \quad \text{and} \quad |x_2 - y_2| \leq |x_1 - y_2|.$$

Since $f(x)$ is monotonically increasing, we can apply the monotonicity of $f$ to the inequalities above:

$$f(|x_1 - y_1|) \leq f(|x_2 - y_1|) \quad \text{and} \quad f(|x_2 - y_2|) \leq f(|x_1 - y_2|).$$

Therefore, we have:

$$f(|x_1 - y_1|) + f(|x_2 - y_2|) \leq f(|x_1 - y_2|) + f(|x_2 - y_1|).$$

This satisfies the required inequality for Case I.

**Case II:** $y_1 \geq \frac{x_1 + x_2}{2}$ or $y_2 \leq \frac{x_1 + x_2}{2}$

Without loss of generality, assume $y_2 \leq \frac{x_1 + x_2}{2}$. In this case, we have the following relations:

$$|x_1 - y_1|, |x_2 - y_2| \leq |x_2 - y_1|,$$

and

$$|x_1 - y_1| + |x_2 - y_2| \leq |x_1 - y_2| + |x_2 - y_1|.$$

Since $f(x)$ is monotonically increasing and convex, we have:

$$f(|x_1 - y_1|) + f(|x_2 - y_2|) \leq f(|x_2 - y_1|) + f(\max\{|x_1 - y_1| + |x_2 - y_2| - |x_2 - y_1|, 0\})$$
$$\leq f(|x_2 - y_1|) + f(|x_1 - y_2|).$$

The first inequality follows from the convexity of $f$, and the second inequality follows from the monotonicity of $f$ and the earlier established relation. This satisfies the required inequality for Case II.

$\square$

Then, by applying Lemma F.1, we obtain:

**Theorem F.2** (Optimal Bipartite Matching for Monotonically Increasing and Convex Loss Functions)**.** *Let* $A = \{a_1, a_2, \ldots, a_n\}$ *and* $B = \{b_1, b_2, \ldots, b_n\}$ *be two sets of real numbers, with* $b_1 \leq b_2 \leq \cdots \leq b_n$. *Let* $f(x)$ *be a monotonically increasing and convex function, and define the loss function for a matching between* $A$ *and* $B$ *as:*

$$L(\sigma) = \sum_{i=1}^{n} f\left(|a_{\sigma(i)} - b_i|\right)$$

*where* $\sigma$ *is a permutation of the indices* $\{1, 2, \ldots, n\}$.

*Let* $A' = \{a'_1, a'_2, \ldots, a'_n\}$ *be the sorted version of* $A$, *i.e.,* $a'_1 \leq a'_2 \leq \cdots \leq a'_n$, *and similarly, let* $B' = \{b'_1, b'_2, \ldots, b'_n\}$ *be the sorted version of* $B$, *i.e.,* $b'_1 \leq b'_2 \leq \cdots \leq b'_n$. *Then, the matching where* $a_{\sigma^*(i)} = a'_i$ *minimizes the loss function* $L(\sigma)$, *i.e.:*

$$L(\sigma^*) = \sum_{i=1}^{n} f\left(|a'_i - b'_i|\right) \leq L(\sigma)$$

*for any other permutation* $\sigma$.

*Proof.* Without loss of generality, assume that both sets $A$ and $B$ are sorted, i.e., $a_1 \leq a_2 \leq \cdots \leq a_n$ and $b_1 \leq b_2 \leq \cdots \leq b_n$, and that the matching $\sigma^*$ is the identity permutation, i.e., $\sigma^*(i) = i$. We aim to show that the identity matching $\sigma^*$ minimizes the loss function when compared to any other permutation $\sigma$.

Consider any permutation $\sigma \neq \sigma^*$. Since $\sigma$ differs from $\sigma^*$, there must exist a pair of indices $i < j$ such that $\sigma(i) > \sigma(j)$. In this case, we have $a_{\sigma(j)} \leq a_{\sigma(i)}$ and $b_i \leq b_j$. Now, consider the two terms in the loss function corresponding to the mismatched pairs $(a_{\sigma(i)}, b_i)$ and $(a_{\sigma(j)}, b_j)$. By Lemma F.1, we have:

$$f\left(|a_{\sigma(i)} - b_i|\right) + f\left(|a_{\sigma(j)} - b_j|\right) \geq f\left(|a_{\sigma(j)} - b_i|\right) + f\left(|a_{\sigma(i)} - b_j|\right)$$

This inequality shows that swapping the indices $\sigma(i)$ and $\sigma(j)$ in the matching does not increase the overall loss.

By repeatedly applying this swapping process, we can transform the matching $\sigma$ into $\sigma^*$ without increasing the loss at any step. Therefore, the identity matching $\sigma^*$ is optimal and minimizes the loss function $L$.

$\square$

