# OpenReview forum: "From Human Labels to Literature: Semi-Supervised Learning of NMR Chemical Shifts at Scale"
_ICML.cc/2026/Conference — ICML 2026 regular_

### Official Review · Reviewer_pNG7 · 2026-03-12

**Soundness:** 2
**Presentation:** 2
**Significance:** 2
**Originality:** 2
**Overall Recommendation:** 4
**Confidence:** 1

**Summary:**

This paper introduces a semi-supervised framework for learning NMR chemical shifts from millions of unassigned spectra extracted from the literature. It frames chemical shift prediction from spectra as a permutation-invariant set supervision problem. The authors also curate ShiftDB-Lit, a large-scale dataset comprising millions of literature-extracted NMR spectra, and incorporate experimental solvent information into the learning process. This enables more accurate, solvent-aware, and multi-element chemical shift predictions.

**Compliance With Llm Reviewing Policy:**

Affirmed.

**Final Justification:**

Sorry for the late response. I still maintain that ShiftDB-Lit should not be considered a contribution, as the quality of a filtered dataset is inherently intended to be better than that of the raw dataset. After reading the other reviews, **I increased my score but decreased my confidence**.

**Key Questions For Authors:**

See above

**Limitations:**

This paper does not discuss its limitations.

**Strengths And Weaknesses:**

**Strengths**

- The paper is clearly written and easy to follow, and the open-source data and code are valuable for reproducibility.
- Experimental NMR chemical shifts are inherently solvent-dependent, so incorporating a solvent prior is a good idea.

**Weaknesses**

- The ShiftDB-Lit dataset cannot be considered as a contribution of this paper, as it is simply a filtered version of the original dataset[1] using a systematic three-stage process.

[1] Nmrextractor: leveraging large language models to construct an experimental nmr database from open-source scientific publications

- In Line 209, the authors formulate the weakly-supervised (molecule-level) loss as a bipartite matching loss and obtain the minimum by sorting the predicted and observed shifts and matching them in order. I am curious whether the sorting is done in ascending or descending order, and whether this choice would affect the final result.
- In Lines 250–259, the authors categorize solvents into three groups: (1) CDCl₃ (89.1%), (2) DMSO‑d₆ (5.7%), and (3) other infrequent solvents, and encode them as learnable embeddings. How is this categorical information encoded, and what is the difference among the three category embeddings?
- In the experiments section, the authors should specify which datasets are used for supervised and weakly‑supervised training, along with their respective data ratios.
- Table 2 is confusing. What does the superscript “3” indicate? Why are there so many blank entries? Moreover, the baseline setting is unfair. As the authors point out, the ~60% improvement is largely due to the baseline being evaluated under an OOD test.
- I strongly suggest that the authors include more convincing baselines to demonstrate the effectiveness of the semi‑supervised training framework. For instance, first training with the weakly‑supervised molecule‑level loss and then fine‑tuning with the supervised atom‑level loss would help validate the carefully designed loss function. Additionally, an unsupervised baseline using a common masking strategy would also be a strong baseline.
- I am confused about why 3D molecular conformations are generated in Line 130, as I could not find any experiments that actually make use of this 3D information.

---

> ### Author Rebuttal · Authors · 2026-03-31
>
> Thanks for the thoughtful reviews! We address your comments point by point as follows.
>
> **(W=Weaknesses, Q=Question, Re.=Response)**
>
> > Re.1 (W1—data contribution)
>
> Section 3.1 states our dataset is built from NMRexp, not from NMRextractor (cited in the comment). The upstream corpus is produced by PDF parsing, OCSR, and LLM-based extraction; it is relatively crude and not tailored to chemical-shift prediction. Our contribution on the data side is the task-oriented curation—spectrum-text parsing, multi-stage filtering and cleaning, and 3D conformer generation—yielding a higher-quality, benchmark-ready resource for this problem, rather than a trivial filter of an existing shift-prediction dataset.
>
> We notice NMRextractor is related work. We will include it in Related Work and References.
>
> > Re.2 (W2—sorting direction)
>
> We sort predicted and observed shifts in the **same direction** and pair them index-wise after sorting. Switching from ascending to descending reverses both sequences, so the sum of absolute differences—and hence the loss—is unchanged.
>
> See **Reviewer iTNC's Re.7** for a minimal script (Hungarian & sorting loss). It helps clarify the implementation.
>
> > Re.3 (W3—solvent embeddings)
>
> **Encoding.** Each example carries a discrete solvent category id in {1, 2, 3}. We look up a learnable vector `e_solv` with the same width as the backbone from a standard embedding table with **three rows** (one per category); the same injection scheme applies to every category, as described in Section 3.4. In code: `nn.Embedding(3, D)`.
>
> **Difference among the three embeddings.** They are three **independently learned** vectors of the same shape, trained end-to-end. The only structural difference is which solvents map to which row.
>
> > Re.4 (W4: supervised vs. weakly-supervised data and ratios)
>
> Thanks for the suggestion. We will add a one-sentence pointer in Experiments so readers see the datasets, batch sizes, and λ together.
>
> **Datasets**. Section 4.5 summarizes the datasets used for supervised and weakly‑supervised training; our strongest setting uses **NMRShiftDB2** for supervised training and **ShiftDB-Lit** for weakly supervised training.
>
> **Batching (“ratios” in practice).** As in Methods, we use separate batch sizes B₁ for supervised and B₂ for weakly supervised losses, and we scale the weak term by λ (Lines 200–204). Table 7 reports **B₁=4, B₂=16, λ=16**.
>
> > Re.5 (W5—Table 2)
>
> **Superscript “3”.** It denotes a **footnote** that lists the sources of the reported metrics for each method (bottom of the page).
>
> **Blank entries.** In the original table we prioritized NMRShiftDB2 as the primary benchmark; we have now completed the L_mol baseline columns. Details in **Table A4 (Reviewer 6gyS's Re.1)**.
>
> **OOD test.** We agree the comparison on ShiftDB-Lit is partly OOD-vs-ID. NMRShiftDB2 remains the **primary** benchmark; ShiftDB-Lit highlights how semi-supervised training **expands chemical-space coverage** beyond fully labeled regimes—closer to **deployment settings** where inference-time molecules occupy a much broader chemical space than atom-assigned training data can cover. Following Reviewer wmHV’s suggestion, we additionally perform a **structure-based split** for a fairer OOD evaluation, and report the detailed results in **Table A2 (Reviewer wmHV's Re.1)**.
>
> > Re.6 (W6—training schedules & masking pretraining)
>
> Thanks for the constructive suggestion. We report:
>
> (i) **staged training** vs. joint semi-supervised learning—weakly‑supervised pretraining then supervised finetuning is consistently weakest, while **joint training** remains our simple and competitive default;
>
> (ii) **masking self-supervised pretraining** (a 3D atom and coordinate masking prediction task on 209M conformers [1]) vs. training from scratch, where the w/ vs. w/o Uni-Mol rows show a **minor gain** from pretrained weights. Results will be added to the revision.
>
> [1] Zhou et al. Uni-Mol: A Universal 3D Molecular Representation Learning Framework, ICLR 2023.
>
> - Table A1: Semi-supervised training baselines (MAE)
>
> |||¹H|||¹³C|||
> |-|-|-|-|-|-|-|-|
> |||DB2 L_atom|DB2 L_mol|Lit L_mol|DB2 L_atom|DB2 L_mol|Lit L_mol|
> |Pretrain|Finetune|||||||||
> |Semi (DB2&Lit)|—|**0.171**|**0.149**|0.056|0.927|0.777|0.506|
> |Weak (Lit)|Sup (DB2)|0.174|0.154|0.073|0.934|0.791|0.629|
> |Sup (DB2)|Semi (DB2&Lit)|0.173|0.150|**0.055**|0.918|**0.768**|**0.499**|
> |Semi (DB2&Lit)|Sup (DB2)|0.171|0.150|0.058|**0.918**|0.778|0.534|
> |Uni-Mol initialization||||||||||||
> |w/ Uni-Mol||**0.171**|**0.149**|0.056|**0.927**|**0.777**|**0.506**|
> |w/o Uni-Mol||0.173|0.150|**0.054**|0.932|0.780|0.509|
>
> > Re.7 (W7—3D conformations)
>
> We generate 3D conformations because the backbone is an SE(3)-equivariant Transformer that **takes atomic coordinates as input**. It is stated in Related Work (Lines 75–76). We will clarify this in Methods.
>
> > Re.8 (Limitations)
>
> See **Reviewer 6gyS's Re.8**.

---

> > ### Author Rebuttal · Reviewer_pNG7 · 2026-04-03
> >
> > Thanks for your rebuttal. However, as Table A1 shows, the main performance gains actually seem to come from pretraining, which weakens your contribution. Moreover, I still maintain that the ShiftDB-Lit dataset doesn't really count as a contribution.

---

> > > ### Author Response · Authors · 2026-04-03
> > >
> > > Thank you for your review. We respectfully disagree with your comments.
> > >
> > > &nbsp;
> > >
> > > ## **Response-1 (Table A1)**
> > >
> > > Table A1 shows that semi-supervised training and weakly supervised training that **use ShiftDB-Lit** both yield **significant improvements** over training the original NMRNet alone. Our default **joint training** remains **simple and competitive** and **does not** require a multi-stage schedule.
> > >
> > > Both semi-supervised and weakly supervised settings rely on the **ShiftDB-Lit dataset and our proposed bipartite matching loss**. Together with Table A1, this **supports** our paper’s claims and contributions:
> > >
> > > - training on large-scale **literature spectra** without atom-level assignment is **effective**;
> > >
> > > - our **set-level loss** is **effective and robust**—both under **joint training** and under **pretraining plus finetuning**.
> > >
> > > Therefore, we **do not agree** with the reviewer’s conclusion that Table A1 weakens our contribution.
> > >
> > > &nbsp;
> > >
> > > - Table A1: Semi-supervised training baselines (MAE)
> > >
> > > |||¹H|||¹³C||||
> > > |-|-|-|-|-|-|-|-|-|
> > > |||DB2 L_atom|DB2 L_mol|Lit L_mol|DB2 L_atom|DB2 L_mol|Lit L_mol|Note|
> > > |Pretrain|Finetune||||||||||
> > > |-|Sup (DB2)|0.197|0.176|0.140|1.152|1.014|1.259|Original NMRNet|
> > > |**Semi (DB2&Lit)**|—|**0.171**|**0.149**|0.056|0.927|0.777|0.506|Our default method|
> > > |**Weak (Lit)**|Sup (DB2)|0.174|0.154|0.073|0.934|0.791|0.629|Suggested in your comment|
> > > |Sup (DB2)|**Semi (DB2&Lit)**|0.173|0.150|**0.055**|0.918|**0.768**|**0.499**|Our extra baseline|
> > > |**Semi (DB2&Lit)**|Sup (DB2)|0.171|0.150|0.058|**0.918**|0.778|0.534|Our extra baseline|
> > >
> > > &nbsp;
> > >
> > > ## **Response-2 (Contribution on Data)**
> > >
> > > We respectfully disagree and summarize why ShiftDB-Lit **is a concrete contribution**.
> > >
> > > **(i) Data quality.**
> > >
> > > Literature-extracted corpora inevitably contain graph errors (OCSR), spectrum-parsing errors, and spectrum–structure mismatches. Our multi-stage filters remove a large fraction of problematic entries (**not** a trivial filter). The table below shows the scale of this curation step:
> > >
> > > | Data | Before filtering | After filtering | Retained |
> > > | --- | ---: | ---: | ---: |
> > > | ¹H | 1,667,135 | 898,422 | 53.9% |
> > > | ¹³C | 1,455,670 | 704,373 | 48.4% |
> > >
> > > To **quantify** data quality, we **manually audited 300 random samples** from the **filtered** ShiftDB-Lit. The **MAE** between parsed peaks and reference labels is **~0.026 ppm** (¹H) and **~0.206 ppm** (¹³C)—**consistent with experimental noise** [1] and **substantially lower than** values reported for NMRShiftDB (0.09 ppm for ¹H; 0.51 ppm for ¹³C) [2]. **This supports** the claim that our curation yields **high-quality training signal** rather than a crude scrape.
> > >
> > > [1] Jonas, E. & Kuhn, S. Rapid prediction of NMR spectral properties with quantified uncertainty. J. Cheminform. 11, 50, https://doi.org/10.1186/s13321-019-0374-3 (2019).
> > >
> > > [2] Kuhn, S., Kolshorn, H., Steinbeck, C. & Schlörer, N. Twenty years of nmrshiftdb2: A case study of an open database for analytical chemistry. Magn. Reson. Chem. 62, 74–83, https://doi.org/10.1002/mrc.5418 (2024).
> > >
> > > **(ii) Scale relative to standard supervised data.**
> > >
> > > After curation, ShiftDB-Lit contains **898,422** ¹H and **704,373** ¹³C entries, versus **12,800** ¹H and **26,859** ¹³C in NMRShiftDB2 in our setting—roughly **26–70×** larger by nucleus. This is a **substantial expansion** of available training resources for chemical-shift modeling.
> > >
> > > **(iii) Significance (standardization, empirical gains, and community utility).**
> > >
> > > To our knowledge, ShiftDB-Lit is the **first large-scale, systematically curated literature-derived chemical-shift resource** released in a form suitable for **training and benchmarking** on this task. **Reviewer wmHV (S5, significance)** further notes that NMRShiftDB2 gains (¹H MAE: 0.1972→0.1709; ¹³C MAE: 1.1518→0.9270) are achieved with no architectural changes, cleanly isolating the contribution of data and loss design—which directly speaks to **why ShiftDB-Lit matters empirically**. **Reviewer iTNC** independently emphasizes solvent metadata, heteroatom coverage, and value **as data infrastructure for future research**, beyond methodology alone.
> > >
> > > Taken together, ShiftDB-Lit is a **documented, quality-controlled, large-scale resource** that directly enables the semi-supervised results we report; we believe this satisfies the bar for a dataset contribution in our setting.

---

### Official Review · Reviewer_wmHV · 2026-03-13

**Soundness:** 3
**Presentation:** 3
**Significance:** 3
**Originality:** 3
**Overall Recommendation:** 5
**Confidence:** 3

**Summary:**

This paper proposes a semi-supervised framework for NMR chemical shift prediction that combines a small atom-assigned dataset (NMRShiftDB2, ~13–27k molecules) with ~900k unassigned literature-extracted spectra (ShiftDB-Lit). Learning from unassigned spectra is formulated as permutation-invariant set supervision, where the authors prove that for convex monotonically increasing losses, optimal bipartite matching reduces to a sorting-based pairing, replacing the O(n³) Hungarian algorithm. Built on the SE(3)-equivariant NMRNet architecture, the framework adds solvent conditioning through CLS-token embedding injection, capturing solvent effects as a global contextual bias of chemical shifts. The approach achieves 13–20% MAE reductions on NMRShiftDB2 for ¹H and ¹³C, with larger gains on ShiftDB-Lit and disproportionate improvements for underrepresented solvents like DMSO-d₆. The authors also provide initial heteroatom baselines (¹⁹F, ³¹P, ¹¹B, ²⁹Si). The authors also provide initial heteroatom baselines (¹⁹F, ³¹P, ¹¹B, ²⁹Si). An ablation study shows that training on unassigned spectra alone leads to model collapse: the model learns to produce a distribution of shift sets that match the observed spectrum collectively but assigns them to the wrong atoms, causing molecular-level loss to decrease while atom-level accuracy degrades. This establishes that a moderate amount of atom-assigned data is necessary to anchor per-atom correctness under the sorting-based loss.

**Compliance With Llm Reviewing Policy:**

Affirmed.

**Final Justification:**

The authors presented a compelling and technically sound approach to semi-supervised NMR chemical shift prediction, supported by the introduction of the large-scale ShiftDB-Lit dataset and an elegant sorting-based loss formulation. My initial concerns primarily focused on evaluation methodology—specifically, the need for a scaffold-based split to properly isolate semi-supervised gains from simple distribution coverage, as well as questions regarding solvent embedding granularity and missing baselines. The authors provided a highly detailed and constructive rebuttal, supplying new experimental data (including the requested scaffold split and fine-grained solvent categories) that directly and fully resolved my technical concerns. The method's originality in framing unassigned spectra as a permutation-invariant set supervision problem, combined with the proven empirical significance of the approach, makes this a strong contribution to the field. The rebuttal firmly reinforced my prior assessment, and I confidently maintain my recommendation to Accept, provided the newly provided analyses are incorporated into the final manuscript.

**Key Questions For Authors:**

1) Can you evaluate on a scaffold-split held-out set from ShiftDB-Lit that is OOD for both the baseline and semi-supervised model? This would directly address W1 and W2 by isolating the semi-supervised learning gain from distribution coverage.

2) What is the solvent distribution within the "others" category (5,553 ¹H test molecules, 5,485 ¹³C test molecules), and have you tried finer granularity? The ↓9.5% (¹H) and ↓9.0% (¹³C) improvements for "others" vs. ↓46.8% and ↓17.8% for DMSO-d₆ suggest the single embedding may be underperforming. (Addresses W3.)

3) What are the wall-clock training times for the supervised (50 epochs, batch size 8) vs. semi-supervised (10 epochs, batch sizes 4+16) settings on the reported NVIDIA RTX 4090? (Addresses W6.)

4) For ¹⁹F, the R² of 0.7216 is notably lower than for ³¹P (0.9634) and ¹¹B (0.9406). Can you comment on whether this reflects intrinsic difficulty, data noise, or the broader chemical shift range (−300 to 300 ppm per Table 6)?

**Limitations:**

Yes.

**Strengths And Weaknesses:**

S1 (Soundness). The sorting-based loss equivalence is cleanly motivated and formally proven. The proof in Appendix F proceeds through a convexity/monotonicity lemma (Lemma F.1) and a swapping argument (Theorem F.2) that transforms any permutation into the sorted matching without increasing loss. The conditions required that the loss takes the form l(x,y) = f(|x−y|) where f is monotonically increasing and convex are mild and satisfied by MAE, MSE, and Huber loss, making the result broadly applicable rather than architecture specific.

S2 (Soundness). The ablation study (Table 5) is well-designed. The five training configurations cleanly isolate contributions: Exp. 1 (supervised on NMRShiftDB2 only) yields ¹H MAE of 0.1972 and ¹³C MAE of 1.1518 on the atom-level metric; Exp. 2 (weakly-supervised on ShiftDB-Lit only) yields ¹H MAE of 0.2412 and ¹³C MAE of 1.5214, revealing model collapse; Exp. 4 (supervised + weakly-supervised both on NMRShiftDB2) yields ¹H MAE of 0.2152 and ¹³C MAE of 1.1503, showing no benefit from adding weak supervision on the same labeled data; and Exp. 5 (supervised on NMRShiftDB2 + weakly-supervised on ShiftDB-Lit) achieves the best results at ¹H MAE 0.1709 and ¹³C MAE 0.9270. The authors report the model collapse failure mode (Exp. 2) rather than only presenting best-case results, which strengthens trust in the experimental methodology.

S3 (Soundness). The cross-solvent validation (Table 9) is a particularly careful experiment. For instance, on the CDCl₃/DMSO-d₆ pair (419 molecules), correct solvent incorporation yields ¹H MAE of 0.0832, while incorrect incorporation gives 0.2814 and no incorporation gives 0.1228. The consistent pattern across all six solvent-pair conditions correct tokens yielding lowest error rules out the possibility that solvent conditioning is simply learning a global bias.

S4 (Originality). Framing unassigned NMR spectra as permutation-invariant set supervision is a natural but previously unexploited connection. While sorting-based losses exist in the broader ML literature, proving optimality under the stated conditions for this domain rather than using it as a heuristic is a meaningful contribution.

S5 (Significance). The NMRShiftDB2 improvements (¹H MAE: 0.1972→0.1709; ¹³C MAE: 1.1518→0.9270) are achieved with no architectural changes to NMRNet, cleanly isolating the contribution of data and loss design. The ShiftDB-Lit dataset itself at 898,422 ¹H and 704,373 ¹³C entries is roughly 26–70× larger than NMRShiftDB2 and substantially expands available training resources for the field.

S6 (Presentation). The paper is clearly structured. Figure 1 effectively communicates the atom-level vs. molecule-level supervision distinction. The progression from the general bipartite matching formulation (Eq. 2) to the sorting-based simplification (Eq. 4) is well-paced.

Weaknesses

W1 (Major Soundness). The ShiftDB-Lit test set evaluation conflates two effects. The NMRNet baseline is trained only on NMRShiftDB2, making ShiftDB-Lit out-of-distribution for it, while the semi-supervised model trains on ShiftDB-Lit, making it in-distribution. The reported reductions ¹H MAE from 0.1395 to 0.0559 (↓59.9%) and ¹³C MAE from 1.2591 to 0.5060 (↓59.8%) therefore reflect both the value of semi-supervised learning and simple distribution coverage. The authors acknowledge this (lines 283–293) but do not attempt to disentangle the two contributions for instance, by evaluating on a held-out scaffold split that is OOD for both models. This is fixable: a scaffold-based partition of ShiftDB-Lit that excludes training scaffolds from both models would isolate the semi-supervised learning gain.

W2 (Major Soundness). ShiftDB-Lit uses a random 4:1 train/test split (Section 4.1) rather than a scaffold split. For a dataset of ~1.6M molecules, random splitting almost certainly places structurally similar molecules on both sides of the partition, inflating reported metrics. The NMRShiftDB2 results use a pre-defined benchmark split and are less affected, but the ShiftDB-Lit numbers which show the most dramatic gains (59.9% and 59.8% MAE reductions) and carry the paper's strongest narrative claims are the most vulnerable to this concern. This is also fixable by re-evaluating with a scaffold split.

W3 (Major Soundness). The solvent embedding is limited to three categories: CDCl₃ (89.1% of data, 162,509 test molecules), DMSO-d₆ (5.7%, 11,623 test molecules), and "others" (5,553 test molecules for ¹H). Table 3 shows the catch-all "others" category retains the highest MAE (0.0996 for ¹H, 0.8684 for ¹³C with solvent incorporation), and its improvement over the solvent-agnostic baseline is modest (↓9.5% for ¹H, ↓9.0% for ¹³C) compared to the gains for DMSO-d₆ (↓46.8% for ¹H, ↓17.8% for ¹³C). There is no analysis of (a) whether the "others" embedding does meaningful work beyond what the model learns without it, or (b) whether finer solvent granularity (e.g., 5–10 categories) would improve performance. Given that ShiftDB-Lit presumably contains solvent identity metadata beyond these three bins, the coarse grouping appears to be an underexplored design choice rather than a hard constraint. A targeted ablation comparing 3-bin vs. finer-grained solvent embeddings would address this.

W4 (Minor Soundness). The λ sweep (Figure 3) tests powers of two from 2⁰ to 2⁷, and the optimal value is λ=16 (2⁴). On NMRShiftDB2 (L_atom), ¹H MAE ranges from approximately 0.170 at λ=16 to approximately 0.190 at λ=128, while ¹³C MAE ranges from approximately 0.93 at λ=16 to approximately 1.15 at λ=128. A finer grid around the optimum, or a sensitivity analysis showing how flat or sharp the basin is, would strengthen the claim that the method is not highly sensitive to this hyperparameter. This is straightforward to add.

W5 (Minor Presentation). The heteroatom results (Table 4) report MAE values of 2.2809 ppm (¹⁹F), 1.3099 ppm (³¹P), 0.8287 ppm (¹¹B), and 1.9186 ppm (²⁹Si), with R² values of 0.7216, 0.9634, 0.9406, and 0.8901, respectively. However, these are presented without any baselines or comparison methods, making it difficult to contextualize whether these numbers represent strong or weak performance. Even a simple DFT or HOSE comparison would anchor them. The ¹⁹F R² of 0.7216 in particular seems relatively low and warrants discussion.

W6 (Minor Presentation). Wall-clock training time is not reported. Table 7 shows 10 epochs for semi-supervised vs. 50 for supervised, with batch sizes of 4 (labeled) + 16 (unlabeled) for semi-supervised vs. 8 (labeled only) for supervised, and learning rates of 4e-4 vs. 1e-4 respectively. Since the weakly-supervised data contains 898,422 ¹H and 704,373 ¹³C entries compared to NMRShiftDB2's 12,800 ¹H and 26,859 ¹³C entries (~26-70× larger), the actual computational cost is difficult to assess from epoch counts alone. This matters for practitioners deciding whether to adopt the method.

---

> ### Author Rebuttal · Authors · 2026-03-31
>
> Thanks for the thoughtful reviews! We address your comments point by point as follows.
>
> **(W=Weaknesses, Q=Question, Re.=Response)**
> > Re.1 (W1—OOD vs ID; W2,Q1—scaffold split)
>
> Thanks for the constructive suggestion of **scaffold-based split**. We acknowledge a random split on ShiftDB-Lit can place structurally similar molecules on both sides of the partition. To better disentangle the contributions of data coverage and semi-supervised learning, we stratify DB-Lit test molecules by their maximum Tanimoto similarity (2048-bit Morgan fingerprints) to the union of training structures from DB2 and DB-Lit: **similar(≥0.7), intermediate(0.5-0.7), dissimilar(<0.5)**.
>
> Table A2 summarizes the results. On the **similar** subset, the semi-supervised model gains the most—consistent with narrowing the OOD–ID gap when training better covers the relevant chemical space. On the **dissimilar** subset, the semi-supervised model still clearly outperforms the supervised baseline, but the improvement is less pronounced than on the similar subset.
>
> - Table A2: OOD-vs-ID evaluation
>
> ||dissimilar||intermediate||similar||
> |-|-|-|-|-|-|-|
> ||MAE|RMSE|MAE|RMSE|MAE|RMSE|
> |¹H|-|-|-|-|-|-|
> |Baseline|0.196|0.426|0.148|0.300|0.133|0.266|
> |Semi-supervised|0.121|0.362|0.067|0.211|0.049|0.166|
> |↓|38.5%|15.1%|54.7%|29.7%|63.1%|37.7%|
> |¹³C|-|-|-|-|-|-|
> |Baseline|2.211|5.299|1.419|3.423|1.136|2.503|
> |Semi-supervised|1.392|4.781|0.645|2.862|0.403|1.907|
> |↓|37.0%|9.8%|54.5%|16.4%|64.5%|23.8%|
>
> We will add these detailed results in the revision.
> > Re.2 (W3,Q2—fine-grained solvent embedding)
>
> We perform a finer-grained solvent categorization:
>
> - Table A3: Fine-grained solvent results
>
> ||¹H|||¹³C|||
> |-|-|-|-|-|-|-|
> |Solvent|Num|w/ solv.|w/o solv.|Num|w/ solv.|w/o solv.|
> |CDCl3|162509|.05(↓5%)|.05|126364|.48(↓3%)|.49|
> |DMSO-d6|11623|.07(↓47%)|.12|9026|.68(↓18%)|.82|
> |CD3COCD3|1263|.07(↓46%)|.12|1123|.81(↓22%)|1.05|
> |CD2Cl2|1030|.05(↓10%)|.06|730|.64(↓12%)|.72|
> |C6D6|589|.09(↓60%)|.23|424|.89(↓8%)|.97|
> |CD3CN|422|.08(↓29%)|.11|336|.94(↓21%)|1.20|
> |CD3OD|253|.17(↓26%)|.23|1105|1.11(↓23%)|1.45|
> |THF-d8|38|.12(↓19%)|.15|33|.83(↓10%)|.93|
> |D2O|34|.55(↓3%)|.57|144|1.55(↓9%)|1.71|
> |DMF-d7|20|.08(↓30%)|.12|17|.61(↓30%)|.87|
> |C2D2Cl4|18|.11(↓1%)|.11|10|.59(↓0%)|.59|
> |PhMe-d8|17|.11(↓48%)|.22|12|1.00(↓3%)|1.03|
> |CF3CO2D|11|.40(↓9%)|.44|5|1.70(↓9%)|1.86|
> |pyridine-d5|6|.23(↓3%)|.24|7|2.13(↓-5%)|2.04|
> |CD3CO2D|—|—|—|3|1.55(↓12%)|1.76|
> |Not known|1852|.06(↓1%)|.06|1536|.56(↓2%)|.57|
> |All|176985|.049(↓12%)|.056|140875|.502(↓5%)|.528|
>
> Analysis:
> - A finer-grained solvent categorization **further promotes the prediction** (0.050→0.049 for H and 0.504→0.502 for C).
> - CDCl3 shows relatively limited improvement, due to it **dominates the dataset**. As a result, models without explicit solvent information implicitly learn a bias toward CDCl3-like environments.
> - For more sensitive ¹H shifts, **nonpolar solvents** with similar properties to CDCl3 (CD2Cl2 and C2D2Cl4) show relatively small improvements. In contrast, **polar or hydrogen-bonding solvents** (DMSO-d6, acetone-d6, CD3OD, THF-d8, DMF-d7) and **aromatic solvents** (C6D6 and PhMe-d8) exhibit larger gains, consistent with chemical intuition that stronger solute–solvent interactions more significantly perturb proton chemical shifts.
>
> We will add detailed results and analysis in the revision.
> > Re.3 (W4—λ sensitivity)
>
> We refine the sweep around the previous optimum (λ∈[8,32]). Results in https://anonymous.4open.science/r/NMRNetplusplus-8C70/figure/exp.pdf. Minima are λ≈20(¹H) and λ≈10(¹³C).
>
> The loss landscape is **flat**: on NMRShiftDB2 (L_atom MAE), ¹H stays 0.170–0.172 for λ∈[8,28] and ¹³C stays 0.92–0.94 for λ∈[4,20].
> > Re.4 (W5,Q4—heteroatom baselines)
>
> - HOSE is database-driven and heteronuclear chemical-shift data are scarce, so few HOSE benchmarks exist for these nuclei.
> - DFT depends strongly on the basis set and is expensive; prior work typically reports errors on **small custom test sets**, not at full-corpus scale. Some results of DFT are listed in the table for reference only (MAE/RMSE).
>
> |Method|¹⁹F|³¹P|¹¹B|²⁹Si|
> |-|-|-|-|-|
> |DFT|~5.2/-|-/~7.0|-/~3.5|~5.9/-|
> |ShiftDB-Lit|2.28/8.76|1.31/4.69|0.83/2.86|1.92/5.23|
>
> Overall, DB-Lit test errors are **below** these literature DFT results. ¹⁹F has relatively large errors, possibly because ¹⁹F spans a wider range.
>
> [1]Ukhanev. A Quest for Effective 19F NMR Spectra Modeling. 2025.
>
> [2]Latypov. Quantum chemical calculations of 31P NMR chemical shifts. 2020.
>
> [3]Gao. 11B NMR chemical shift predictions via density functional theory and gauge-including atomic orbital approach. 2019.
>
> [4]Du. 29Si NMR Shielding Calculations Employing Density Functional Theory. 2011.
>
> > Re.5 (W6,Q3—wall-clock training)
>
> On one RTX 4090, supervised takes 2.31h(¹H)/4.58h(¹³C). Semi-supervised takes 24.80h(¹H)/18.34h(¹³C), much larger but acceptable. We also ran supervised training to 100/200 epochs, and validation plateaued, so the 50-epoch supervised run is **not under-trained**.

---

> > ### Author Rebuttal · Reviewer_wmHV · 2026-04-06
> >
> > I thank the authors for their highly detailed and constructive rebuttal. You have directly addressed all of my concerns, particularly through the addition of new experimental data. All of my technical concerns are fully resolved. I expect the authors to include these tables, baselines, and analyses (especially the scaffold split and fine-grained solvent data) in the final revision, as they substantially strengthen the manuscript. I maintain my recommendation to Accept.

---

> > > ### Author Response · Authors · 2026-04-06
> > >
> > > Thanks for acknowledging our work and for your constructive suggestions. The tables, baselines, and analyses (especially the scaffold split and fine-grained solvent data) will be included in the final revision. We truly appreciate your time and effort in reviewing our paper.

---

### Official Review · Reviewer_6gyS · 2026-03-13

**Soundness:** 3
**Presentation:** 4
**Significance:** 4
**Originality:** 2
**Overall Recommendation:** 4
**Confidence:** 4

**Summary:**

The authors propose a weakly-supervised framework to learn an NMR chemical shift predictor from structure using literature extracted weakly labeled NMR spectra and a small amount of strongly labeled (atom level assigned) NMR spectra from NMRShiftDB2. Another contribution is to provide a solution to condition the prediction on the used solvent. Empirical results show performance increase relative to SOTA.

**Compliance With Llm Reviewing Policy:**

Affirmed.

**Final Justification:**

The rebuttal addressed several of my concerns and lead to improved score. The work is a nice application of existing ML techniques to a relevant problem. I in general would advocate for acceptance, the only reason I am not 100% convinced is that the contribution to the ML field outside of the specific application area itself is not clear for me, this is related to lack of technological novelty.

**Key Questions For Authors:**

- Please provide error bars for your MAE and RMSE values, and provide the missing values in Table 2.
- In general the term “unsupervised” is used many times in the paper, but actually the literature data is not unsupervised as there is a label, a set label. As properly named other places, this is a weakly supervised setting. To avoid confusion please change the occurences of "unsupervised" to "weakly-supervised" in the text. Also semi-supervised can be changed to weakly-supervised. (Except for the title, that you cannot change anymore.)
- It seems you ignore multiplicity of 1H NMR peaks altogether. This is an easy to use , and valuable information to restrict your possible permutations. Did you tried using it?
- On Figure 3 the model collaps argument about the red line in case of 1H seems a bit fragile if we see that 13C line do not show this behaviour. There is nothing mentioned about this in the text. Please elaborate what could be the possible reason?
- How many conformers you generate and use as input?

**Minor:** using $\hat{s}$ as ground truth and $s$ as the prediction goes against the traditional use, where hat-variable denotes the estimate, would be more appropriate to switch the two notations.

**Limitations:**

No explicit discussion of limitations are provided. One that come into my mind is the difficulty to guess the dominant conformer(s) in the sample, as that can depend also on the solvent for example, furthermore there can be an ensemble of conformers contributing to the signal.

**Strengths And Weaknesses:**

Strengths:
- Predicting NMR specra is a relevant problem in multiple different application of analytical chemistry, including organic chemistry research, pharmaceutical industry (assessment of impurities at production), identifying environmental pollutants, and so on.
- The suggested method utilizes a vast amount of untaped resource of literature data.
- It addresses the problem of solvent effects

Weaknesses:
- No error bars are provided in any of the result tables, It is very hard to tell if the differences are significant or not.
- Many element of Table 2 left unfilled but I fail to see the reason why they cannot be computed.
Any prediction method can be applied to the structures in the ShiftDB-Lit database, and the $L_{mol}$ loss can be computed on the prediction. I may missing something (in that case please help me here), but I see no reason why the missing values cannot be computed in Table 2.
- The idea of using a set based supervision loss is not particularly novel.

**Addressing the evaluation problem (adding error bars, providing all losses in Table 2 that can be compute) would automatically result +1 in my overall score.**

---

> ### Author Rebuttal · Authors · 2026-03-31
>
> Thanks for the thoughtful reviews! We address your comments point by point as follows.
>
> **(W=Weaknesses, Q=Question, Re.=Response)**
> > Re.1 (W1,Q1—error bars; W2—Table 2 blanks)
>
> Thanks for helping us make Table 2 more complete.
>
> **Error bars.** We report **mean ± standard deviation** over five random seeds for semi-supervised NMRNet; variation is small, indicating stable training.
>
> **Missing cells.** These metrics are well-defined and computable. We have reproduced all baselines and filled the missing cells.
>
> - Table A4: Completed Table 2
>
> ¹H
> |Method|DB2 L_atom||DB2 L_mol||DB-Lit L_mol||
> |-|-|-|-|-|-|-|
> ||MAE|RMSE|MAE|RMSE|MAE|RMSE|
> |HOSE|0.310|0.659|0.277|0.556|0.216|0.393|
> |GCN|0.242|0.549|0.215|0.455|0.171|0.340|
> |FCG|0.225|0.491|0.204|0.420|0.156|0.299|
> |SGNN|0.215|0.487|0.192|0.403|0.150|0.294|
> |GT-NMR|(0.158)*|(0.293)*|—|—|—|—|
> |NMRNet Baseline|0.197|0.456|0.176|0.390|0.140|0.279|
> |NMRNet Semi-supervised|0.172±0.001|0.438±0.003|0.150±0.001|0.365±0.002|0.055±0.001|0.184±0.001|
> |↓|12.9%|4.1%|15.0%|6.4%|60.7%|34.2%|
>
> *(GT-NMR predicts only hydrogens bonded to carbon; full-atom comparison is therefore not applicable)
>
> ¹³C
> |Method|DB2 L_atom||DB2 L_mol||DB-Lit L_mol||
> |-|-|-|-|-|-|-|
> ||MAE|RMSE|MAE|RMSE|MAE|RMSE|
> |HOSE|2.580|4.850|2.309|4.190|2.375|4.415|
> |GCN|1.304|2.510|1.157|2.191|1.255|2.951|
> |FCG|1.359|2.349|1.219|2.063|1.262|2.890|
> |SGNN|1.261|2.210|1.121|1.914|1.201|2.877|
> |GT-NMR|1.165|2.143|1.039|1.865|1.108|2.807|
> |NMRNet Baseline|1.152|2.140|1.014|1.851|1.259|2.921|
> |NMRNet Semi-supervised|0.929±0.002|1.927±0.012|0.778±0.001|1.577±0.013|0.507±0.001|2.350±0.001|
> |↓|19.4%|9.9%|23.3%|14.8%|59.8%|19.5%|
> > Re.2 (W3—novelty of set loss)
>
> We agree that set-based supervision is **not new as a general ML idea**. Our contribution is not to propose the abstract notion of “train on unordered sets,” but to **instantiate and scale** it for chemical-shift prediction from literature spectra where atom–peak assignment is missing.
>
> Our central claim is that training on large-scale literature spectra without atom-level assignment is effective: on NMRShiftDB2, the gains exceed those obtained from recent architectural changes alone. We emphasize a shift from model-centric to **data-centric AI** for scientific prediction, and view this work as a step toward scalable **scientific data infrastructure**.
> > Re.3 (Q2—terminology)
>
> Agree with the opinion. We use "semi-supervised" to refer to the training with both atom-level and set-level data, to distinguish it from the "weakly-supervised" setting that only uses set-level data. We will change the occurrences of "unsupervised" to "weakly-supervised" in the revision.
> > Re.4 (Q3—multiplicity constraint)
>
> We tried restricting permutations with multiplicity for ¹H, but it yielded no gain. Two reasons: (i) over 40% of ¹H peaks are **labeled "m" (multiplet)**, and occasional wrong labels can incorrectly exclude valid permutations; (ii) the constraint assumes **reliable multiplicity-from-structure prediction**, which the current rule-based method does not yet provide. Under these conditions, multiplicity pruning has limited benefits.
>
> We still consider the direction promising—e.g., stronger multiplicity predictors or higher-quality data (including raw spectra) may help.
> > Re.5 (Q4—λ vs L_mol)
>
> In Fig.3, increasing λ drives ShiftDB-Lit L_mol down almost monotonically for ¹H but yields a U-shaped curve for ¹³C. We interpret this as: for ¹H, large λ favors fitting ShiftDB-Lit's L_mol over atom-level accuracy; for ¹³C, L_mol does not keep improving with λ alone, and a small amount of atom-level supervision can help.
>
> We attribute this partly to ¹H shifts spanning **a narrow range**: many alternative peak-to-atom matchings can yield similarly low L_mol, so that objective is **easier to overfit**; ¹³C spans a wider range, so low L_mol depends more on **correct assignments**, yielding a U-shaped curve.
>
> We will expand the discussion in the revision.
> > Re.6 (Q5—conformers)
>
> We use **one** conformer per molecule for scalability. Tested on 1000 molecules, averaging predictions over 10 random conformers changed MAE by **~0.01 ppm(¹H) / ~0.1 ppm(¹³C)** relative to a single conformer, which is negligible. So we keep single-conformer training and evaluation, and multi-conformer sampling may still help for highly flexible molecules during inference.
> > Re.7 (notations)
>
> Thanks for catching this. We will adopt the standard convention: $\{s_i\}$ for ground truth and $\{\hat{s}_i\}$ for predictions.
> > Re.8 (Limitations)
>
> A Limitations section will be added in the revision to discuss this.
> We agree that determining the dominant conformer in solution is not trivial. It can depend on solvent and other experimental factors.
> - The current method uses a simple solvent embedding method, **more complex solvent models** can be considered, e.g. consider solvent environment in conformer generation.
> - **Multi-conformer input** prediction is also worth exploring, especially for highly flexible molecules.

---

> > ### Author Rebuttal · Reviewer_6gyS · 2026-04-02
> >
> > Thank you for the detailed answers. I will increase my score.

---

> > > ### Author Response · Authors · 2026-04-06
> > >
> > > Thanks for your acknowledge of our work and for raising score. Thanks again for your time and effort in reviewing our paper!

---

### Official Review · Reviewer_iTNC · 2026-03-15

**Soundness:** 3
**Presentation:** 3
**Significance:** 3
**Originality:** 3
**Overall Recommendation:** 4
**Confidence:** 4

**Summary:**

The authors claim to address an important concept: overcoming the data bottleneck in NMR chemical shift prediction by combining a small atom-assigned dataset with millions of literature-extracted but unassigned spectra. This paper attempts to address the area of weakly/semi-supervised learning for scientific data where exact labels are expensive but weak structural signals are abundant. The core idea is to formulate unassigned NMR supervision as a permutation-invariant set prediction problem, then show that under common regression losses the matching objective reduces to a sorting-based loss, making large-scale training practical. The paper also introduces ShiftDB-Lit, a large literature-derived dataset with solvent metadata, and studies solvent-aware conditioning as well as heteroatom shift prediction. Empirically, the method improves over the NMRNet baseline on both NMRShiftDB2 and ShiftDB-Lit, with especially large gains on the literature-scale benchmark.

**Compliance With Llm Reviewing Policy:**

Affirmed.

**Final Justification:**

My concerns regarding the manuscript are fully resolved, according to authors’ responses and explanations.

**Key Questions For Authors:**

1. Can the authors quantify the noise level of ShiftDB-Lit more directly, for example by manually auditing a random subset of extracted examples?
2. How much of the gain comes from the sorting-based weak loss itself versus simply exposing the model to much broader chemical coverage?
3. Did the authors deduplicate near-identical molecules or control for overlap between literature-derived molecules and benchmark molecules?
4. Could the authors compare against a Hungarian-loss implementation directly on a smaller subset to verify that the sorting surrogate is empirically equivalent in practice, not only theoretically?
5. For solvent modeling, what happens if the most common additional solvents are modeled separately rather than merged into “others”?

**Limitations:**

The paper would be stronger with one extra section on data auditing: extraction quality, solvent label normalization, duplicates, and error examples.
It would also help to separate the claims more clearly:
- “semi-supervised objective is effective,”
- “large-scale literature data improves coverage,”
- “solvent conditioning adds value.”

Right now these are somewhat intertwined.
I would also like a clearer comparison to other possible permutation-invariant objectives and perhaps a discussion of when the sorting reduction fails if the loss assumptions are violated.

**Strengths And Weaknesses:**

## Strengths:
The problem is important and well motivated. Existing ML methods depend on scarce atom-level assignments, while literature contains much larger amounts of unassigned spectra. Framing this as weak supervision is natural and potentially impactful beyond NMR.

The technical formulation is clean. The paper defines a bipartite matching objective over predicted and observed shifts, then argues that for losses of the form $l(x,y)=f(|x-y|)$with monotone convex $f$, the optimum is achieved by sorting both sets and matching in order. This is a nice simplification that converts a combinatorial assignment problem into a deterministic training loss.

The dataset contribution is substantial. ShiftDB-Lit is much larger than NMRShiftDB2 and includes solvent information plus several heteroatoms. That makes the work useful not only methodologically but also as infrastructure for future research.

The experimental gains are strong. On NMRShiftDB2, semi-supervised training improves the NMRNet baseline for both ^1H and ^13C; on ShiftDB-Lit, the gains are much larger. The solvent-conditioning experiments are also interesting, especially the strong benefit on underrepresented solvents such as DMSO-d6.

I also appreciate the ablation in Table 5 showing an important negative result: weak supervision alone collapses, but works well when anchored by labeled data. That gives the paper a more credible and nuanced story than simply “more data helps.”

## Weaknesses / concerns

My main concern is evaluation fairness and interpretation. The paper emphasizes large gains on ShiftDB-Lit, but the baseline is trained only on NMRShiftDB2 while the proposed model leverages ShiftDB-Lit during training, so the comparison is partly OOD-vs-ID rather than purely method-vs-method. The paper does acknowledge this, but the framing should be more careful.

A second concern is novelty level. The application is valuable, and the sorting reduction is elegant, but the overall method is still a relatively direct semi-supervised extension of an existing backbone rather than a fundamentally new model family. The strongest novelty is really the formulation plus the scale of the data resource.

A third concern is data quality / noise robustness. The literature-extracted dataset is large, but inevitably noisy. The paper describes filtering procedures, which is good, but the main text does not quantify error rates from extraction, OCSR, parsing, solvent normalization, or duplicate handling. Since the paper’s central claim relies on learning from noisy literature data, this deserves more explicit auditing.

The solvent modeling is promising but still fairly coarse. Solvents are grouped into three categories, with “others” collapsed into one embedding. That is understandable for data imbalance reasons, but it limits chemical interpretability and may hide meaningful solvent-specific behavior.

Finally, the paper would benefit from stronger baselines in the weak-supervision setting. Most comparisons are against older supervised predictors or the NMRNet baseline. There is less discussion of alternative set-prediction / permutation-invariant training objectives or semi-supervised baselines beyond the chosen formulation.

---

> ### Author Rebuttal · Authors · 2026-03-31
>
> Thanks for the thoughtful reviews! We address your comments point by point as follows.
> **(W=Weaknesses, Q=Question, Re.=Response)**
>
> > Re.1 (W1,Q2—fairness/interpretation)
>
> Agree the comparison on ShiftDB-Lit is partly OOD-vs-ID, not a pure method-only fair test. NMRShiftDB2 remains the **primary** benchmark; ShiftDB-Lit highlights how semi-supervised training **expands chemical-space coverage** beyond fully labeled regimes—closer to **deployment settings** where inference-time molecules occupy a much broader chemical space than atom-assigned training data can cover.
>
> The ablation in Table 5 shows that adding the weak loss alone does not improve performance on the same atom-assigned training data; its role is to enable learning from a much larger assignment-free corpus. Thus the weak-loss objective and data coverage are **coupled** contributions. Following Reviewer wmHV’s suggestion, we additionally run a **structure-based split** for a fairer OOD evaluation; details are in **Table A2 (Reviewer wmHV's Re.1)**.
>
> > Re.2 (W2—novelty)
>
> We agree that the strongest novelty lies in the learning formulation and data resource rather than in a fundamentally new architecture. Our central claim is that training on large-scale literature spectra without atom-level assignment is effective: on NMRShiftDB2, the gains exceed those obtained from recent architectural changes (Table 2).
>
> We emphasize a shift from model-centric to **data-centric AI** for scientific machine learning, and view this work as a step toward scalable scientific data infrastructure and large-scale learning from weakly labeled scientific data.
>
> > Re.3 (W3,Q1—data quality & noise level)
>
> Thanks for this important point. The literature-extracted dataset is inevitably noisy. Our multi-step filtering process is designed to mitigate this. We **manually audited** 300 random samples and found MAE **~0.026 ppm(¹H)/0.206 ppm(¹³C)**, consistent with experimental noise expectations, so data quality is ensured.
>
> The “noisy” in Line 373 refers to the “pseudo-labels” for weak supervision, which introduces uncertainty. The hyperparameter λ balances this by adjusting the relative weight of atom-level versus molecular-level objectives. We will clarify this in the revision.
>
> > Re.4 (W4,Q5—fine-grained solvent categories)
>
> Thanks for raising this point. The original solvent grouping was coarse; we added a finer-grained setup with 15 solvent labels, with detailed experiments and analysis in **Table A3 (Reviewer wmHV's Re.2)**.
>
> > Re.5 (W5—weak-supervision baselines)
>
> **Alternative set-prediction objectives.**
> We use bipartite matching, which is standard in machine learning, for the one-to-one pairing of active nuclei with chemical shifts. When peak assignments are fixed, the objective **reduces to a supervised per-atom loss**. This family includes common set-prediction objectives—for example, **Wasserstein-1 distance (OT)** is the MAE version of bipartite matching loss in the 1D discrete setting. We also compared MAE, MSE, and Huber as the pairwise regression term and observed similar evaluation performance.
>
> **Semi-supervised baselines.**
> We compare **joint semi-supervised training** vs. **pretrain–finetune baseline** under several data combinations. Detailed results are in **Table A1 (Reviewer pNG7's Re.6)**.
>
> > Re.6 (Q3—deduplication and benchmark overlap)
>
> We **deduplicate exact structures** and do not merge **near-identical** molecules with different structure. We confirm that literature-derived entries **do not overlap** the benchmark data, and that ShiftDB-Lit has **no train–test overlap**.
>
> > Re.7 (Q4—numerical equivalence of losses)
>
> Thanks for the suggestion. As a lightweight check, we drew 10000 random instances and compared the outputs of the two implementations. In every trial the two losses **agreed within numerical tolerance**. Code is given below.
>
> ```python
> import numpy as np
> from scipy.optimize import linear_sum_assignment
> rng = np.random.default_rng(0)
> for _ in range(10_000):
>     n = rng.integers(2, 50)
>     set1 = rng.random(n)
>     set2 = rng.random(n)
>     cost = np.abs(set1[:,None] - set2[None,:])
>     r, c = linear_sum_assignment(cost)
>     loss_hungarian = cost[r,c].sum()
>     loss_sort = np.abs(np.sort(set1) - np.sort(set2)).sum()
>     assert abs(loss_hungarian - loss_sort) < 1e-8
> ```
>
> > Re.8 (Limitations)
>
> Thanks for these constructive suggestions. The **additional experiments** added throughout our responses already address several of these points: (1) **semi-supervised baselines**; (2) a **structure-based split** for a fairer OOD evaluation; (3) **fine-grained solvent categories**.
>
> **When the sorting reduction fails.** The sorting reduction fails when the elements of the set fall into the **non-convex region** of the loss function $l(x)$. The commonly used loss functions are convex, because a larger margin penalty is tend to be added for larger errors.
>
> We will add these experiments and discussion in the revision to separate claims more clearly.

---

> > ### Author Rebuttal · Reviewer_iTNC · 2026-04-03
> >
> > My concerns regarding the manuscript are fully resolved, according to authors’ responses and explanations.

---

> > > ### Author Response · Authors · 2026-04-06
> > >
> > > Thanks for your acknowledge of our work. Thanks again for your time and effort in reviewing our paper!

---

### Decision · Program_Chairs · 2026-04-30

**Decision:**

Accept (regular)

**Comment:**

There is overall consensus that this paper contributes a non-trivial dataset which would enable the use of AI technology for understanding NMR chemical shifts. This should be of interest to AI for Science audience.